# OCP–FRP protein complex topologies suggest a mechanism for controlling high light tolerance in cyanobacteria

Nikolai N. Sluchanko [1,2], Yury B. Slonimskiy[1,3], Evgeny A. Shirshin [4], Marcus Moldenhauer[5], Thomas Friedrich [5] & Eugene G. Maksimov [2]

In cyanobacteria, high light photoactivates the orange carotenoid protein (OCP) that binds to antennae complexes, dissipating energy and preventing the destruction of the photosynthetic apparatus. At low light, OCP is efficiently deactivated by a poorly understood action of the dimeric fluorescence recovery protein (FRP). Here, we engineer FRP variants with defined oligomeric states and scrutinize their functional interaction with OCP. Complemented by disulfide trapping and chemical crosslinking, structural analysis in solution reveals the topology of metastable complexes of OCP and the FRP scaffold with different stoichiometries. Unable to tightly bind monomeric FRP, photoactivated OCP recruits dimeric FRP, which subsequently monomerizes giving 1:1 complexes. This could be facilitated by a transient OCP–2FRP–OCP complex formed via the two FRP head domains, significantly improving FRP efficiency at elevated OCP levels. By identifying key molecular interfaces, our findings may inspire the design of optically triggered systems transducing light signals into protein–protein interactions.

[1] A.N. Bach Institute of Biochemistry, Federal Research Center of Biotechnology of the Russian Academy of Sciences, Leninskiy prospect 33, building 1, 119071 Moscow, Russian Federation. [2] M.V. Lomonosov Moscow State University, Department of Biophysics, Faculty of Biology, Leninskie gory 1, building 24, 119234 Moscow, Russian Federation. [3] M.V. Lomonosov Moscow State University, Department of Biochemistry, Faculty of Biology, Leninskie gory 1, building 12, 119234 Moscow, Russian Federation. [4] M.V. Lomonosov Moscow State University, Department of Quantum Electronics, Faculty of Physics, Leninskie gory 1, building 62, 119991 Moscow, Russian Federation. [5] Technical University of Berlin, Institute of Chemistry PC 14, Straße des 17. Juni 135, D-10623 Berlin, Germany. Correspondence and requests for materials should be addressed to N.N.S. (email: nikolai.sluchanko@mail.ru)

Photosynthesis is a pivotal process that changed our planet dramatically during ~2.5 gigayears of its evolution[1,2]. Its efficiency highly depends on the ability of photosynthetic organisms to tolerate seriously different levels of insolation. Photoprotection systems allow plants, algae and cyanobacteria to survive and prosper under high light conditions when the risk of reactive oxygen species production and destruction of the photosynthetic apparatus is increased. The common goal notwithstanding, molecular mechanisms of such adaptation differ significantly[3,4]. The photoprotection mechanisms of cyanobacteria are dictated by the specific nature of their water-soluble light-harvesting antenna complexes, phycobilisomes (PBs)[5,6], which gather light in a wide spectral range and transfer excitation energy to the photosystems. To control this energy flow, cyanobacteria uniquely rely on the functioning of the photoactive orange carotenoid protein (OCP). OCP combines the functions of a light intensity sensor and a trigger of the process of quenching of the excessive PBs excitation[7,8].

OCP was first described in 1981 as an orphan carotenoid-binding protein[9], and its role in photoprotection was unraveled rather recently[7,8,10,11]. The first crystal structure of OCP[12] had been obtained before the function of the protein was understood. OCP is composed of the N-terminal (NTD) and C-terminal (CTD) domains, forming a central channel occupied by a single non-covalently bound xanthophyll molecule, and is stabilized by interactions across the domain interface and the attachment of the N-terminal extension (NTE, residues 1–20) to the β-sheet surface on the CTD. The role of the NTE in OCP photoactivation is widely discussed[13–15]. Blue-green light absorption causes a reversible transition of OCP from the basal orange (OCP$^O$) form with compact structure to the red (OCP$^R$) form with the NTE detached and separated protein domains[16–18]. Only OCP$^R$ is thought to quench PBs fluorescence by directly interacting with the PBs core[8,19–22]. This photoactivated OCP form is metastable but can be mimicked by mutation of the conserved Tyr/Trp residues coordinating the ketocarotenoid, which leads to destabilization of the compact protein structure and separation of the domains, such as in OCP$^{W288A}$ [23,24] and OCP$^{Y201A/W288A}$ (hereafter, OCP$^{AA}$)[15,25] variants.

The process of OCP$^R$ relaxation to OCP$^O$ spontaneously happens in the dark, but is dramatically accelerated by the action of the recently discovered 14 kDa fluorescence recovery protein (FRP)[26], which terminates photoprotection and recovers PBs fluorescence[22,27]. PBs, OCP, and FRP represent the three principal components of the cyanobacterial photoprotection mechanism that is functional also in vitro[22]. Despite the efforts of several laboratories, the whole chain of molecular events associated with the OCP-mediated photoprotection mechanism remains poorly understood, mainly due to the remarkable metastability of the photoactivated OCP$^R$ state and the dynamic and transient nature of its complexes with PBs and FRP[22].

FRP crystallizes as an α-helical protein[28,29] forming stable dimeric conformations in solution[24,25,30,31]. Having a rather low affinity to OCP$^O$ ($K_d$ ~35 μM), FRP tightly interacts with OCP$^R$ and its analogs with separated domains ($K_d$ ~1–3 μM)[24,32]. Selective interaction with OCP lacking the NTE, i.e., the ΔNTE mutant, (submicromolar $K_d$)[30], and with individual CTD, but not individual NTD[25,33], implied that the crucial FRP-binding site is located on the CTD, although the possibility of secondary site(s) was also proposed[24,30,34]. Many observations suggested FRP monomerization upon its interaction with various OCP forms[24,25,30,32], however, the necessity and role of this process was unclear[35,36]. Intriguingly, low-homology FRP from Anabaena variabilis and Arthrospira maxima demonstrated the ability to perform on OCP from Synechocystis sp. PCC 6803, but formed complexes by distinct stoichiometries[25]. This suggested

that the FRP mechanism is rather universal across cyanobacterial species;[25] however, the intermediates of the OCP–FRP interaction and the topology of their complexes remained largely unknown.

To provide mechanistic insight, we engineered unique mutants of Synechocystis FRP tentatively representing its constitutively monomeric and dimeric forms, and examined their properties by an alloy of complementary biochemical, optical and structural biology methods. The expected oligomeric states of the mutants were confirmed, that allowed studying the FRP mechanism in unprecedented detail. A back-to-back comparison of the properties of the dissociable wild-type FRP dimer, its monomeric mutant form, and the disulfide-trapped dimeric variant permits an explanation of different stoichiometries (1:1, 1:2, and newly found 2:2) and topology of the otherwise kinetically unstable OCP–FRP complexes. Chemical crosslinking, disulfide trapping and small-angle X-ray scattering (SAXS) data suggest that complexes with different stoichiometry likely represent intermediates of the OCP–FRP interaction. The unraveled molecular interfaces suggest the scaffolding action of the negatively charged extended region of FRP facilitating re-combination of OCP domains with complementary clusters of the opposite charge, providing a platform for the development of innovative optically triggered systems. The proposed dissociative mechanism may substantially improve FRP efficiency in accelerating OCP$^R$–OCP$^O$ back-conversion, especially at elevated levels of photoactivated OCP, which is confirmed by functional tests and biophysical modeling, thereby reconciling several apparently contradictory observations.

## Results

**Design of the monomeric and dimeric FRPs.** The dimeric state of the prototypical Synechocystis FRP and two of its homologs from Anabaena and Arthrospira was shown by size-exclusion chromatography (SEC)[24,25] and the common dimeric conformation in solution was established by SAXS[25], permitting manipulations of the oligomeric state (Fig. 1a). To create a dimerization-deficient FRP, we introduced an L49E mutation into the dimer interface, which would cause its point destabilization (Fig. 1b). Alternatively, we introduced pairs of adjacent Cys residues in the interface region so that formation of disulfide bridges would covalently fix FRP dimers. It was necessary to pick residues separated by ~4–8 Å between their Cβ atoms[37]. Taking into account potential dynamics of FRP dimers, upon fixation of the dimeric interface, we wanted to prevent any sliding and partial detachment of protein chains. To achieve this, we chose almost exclusive positions in the FRP structure, namely L33 and I43, which simultaneously satisfied all the requirements. Importantly, the Cβ atoms of L33 and I43 in each of the two sides of the antiparallel FRP dimer are separated by ~6.5 Å and I43 is located in a more flexible loop region, increasing the chances of disulfide bond formation between the side chains of C33 and C43 upon L33C/I43C (FRPcc) mutation (Fig. 1c).

Both putatively monomeric (L49E) and dimeric (FRPcc) mutants were produced recombinantly and purified to homogeneity under reducing conditions. The decreased hydrodynamic radius and at least partial monomerization of the L49E variant were confirmed by the results of native polyacrylamide gel-electrophoresis (PAGE) showing similar mobility of the wild-type FRP (FRPwt) and FRPcc and the downward shift of L49E (Fig. 1d).

The efficiency of FRPcc oxidation was optimized (Supplementary Fig. 1). Among several oxidation schemes, dialysis against a 1 mM mixture of reduced and oxidized glutathione (GSH/GSSG) resulted in >95% formation of –S–S– crosslinked FRP dimers (oxFRPcc, see Supplementary Fig. 1b), which were rather stable to reduction. Under the conditions used, crosslinking of FRPwt by

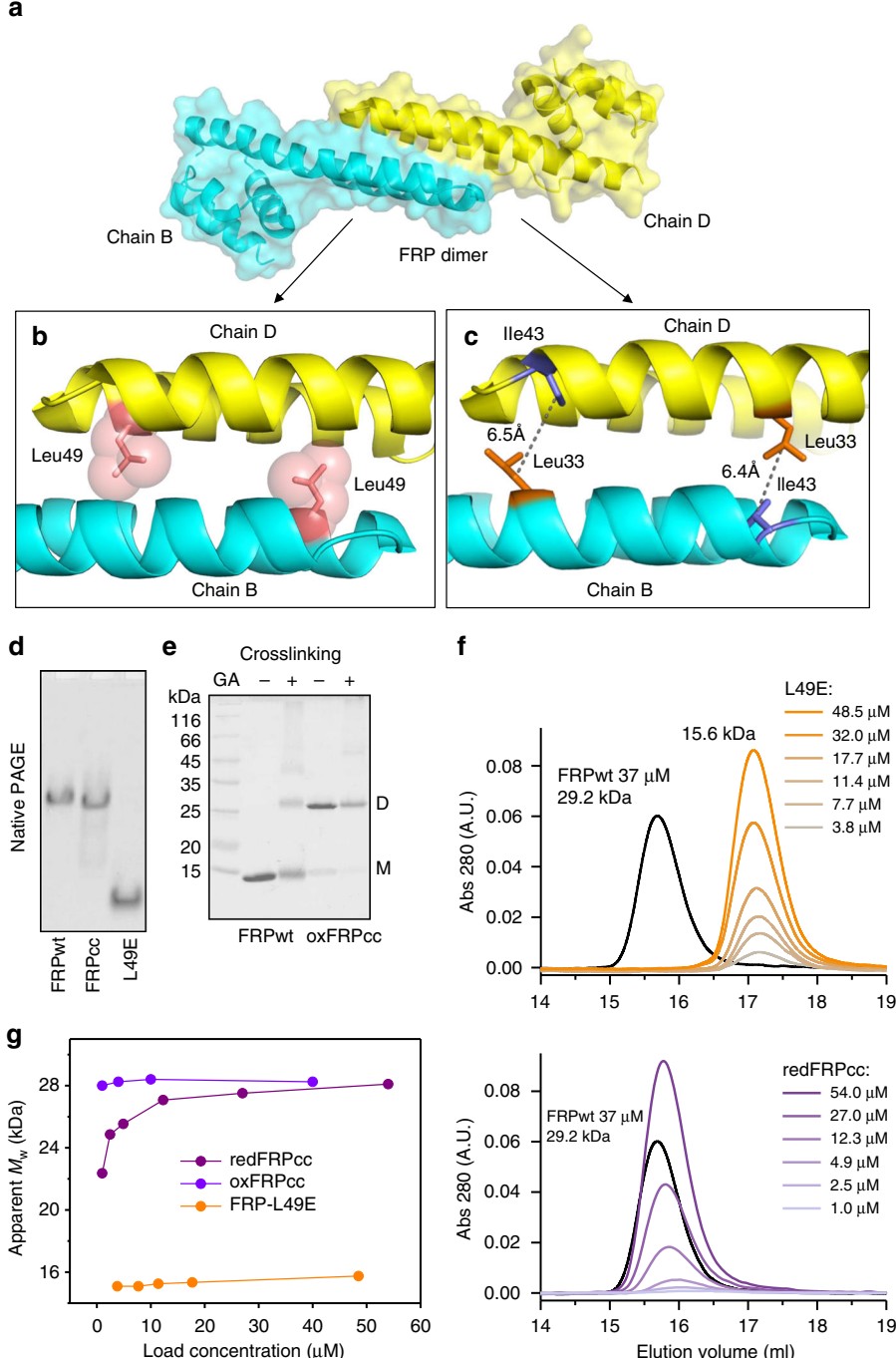

**Fig. 1** FRP mutants with the predefined oligomeric structure. **a** Overall view on the 4JDX structure of the *Synechocystis* FRP dimer with two subunits colored by yellow and cyan. **b** Close-up of the subunit interface showing positions of L49 residues (salmon sticks and semitransparent spheres) mutated to Glu to provoke dimer dissociation. **c** Close-up of the subunit interface showing positions of L33 (orange sticks) and I43 (slate sticks) residues as optimal candidates (Cβ atoms separated by ~6.5 Å) for the intersubunit disulfide crosslinking. Analysis of the quarternary structure of the engineered FRP mutants using native PAGE (**d**) and chemical crosslinking followed by SDS-PAGE (**e**). FRPwt and oxFRPcc were crosslinked in the presence of GA (+ lanes); control samples (− lanes) did not include GA. **f** Analytical SEC on a Superdex 200 Increase 10/300 column of the engineered FRP mutants at different FRP concentrations (indicated in μM per monomer) under reducing conditions. **g** The dependence of the apparent $M_W$ for the FRP-L49E, oxFRPcc, and redFRPcc on loaded protein concentration as calculated from column calibration

glutaraldehyde (GA) produced mainly dimeric species, in agreement with previous work[24]; almost no higher order oligomers were formed by dimeric oxFRPcc (Fig. 1e).

On analytical SEC, the L49E mutant eluted as 15.6 kDa species with invariant peak position over a range of protein concentrations (Fig. 1f), suggesting its monomeric state (calculated $M_W$ 14.1 kDa). FRPwt showed the dimeric peak with $M_W$ ~29 kDa

(Fig. 1f). Under reducing conditions, at high protein concentration loaded on the column (>10 μM), FRPcc (redFRPcc) eluted as dimeric species but showed gradual decrease of the apparent $M_W$ upon manifold dilution (Fig. 1f, g), undergoing partial dimer dissociation, like FRPwt[24]. In contrast, we observed almost unchanged position of the FRPcc peak if the protein was pre-oxidized (Fig. 1g). This strongly indicated covalent fixation of the

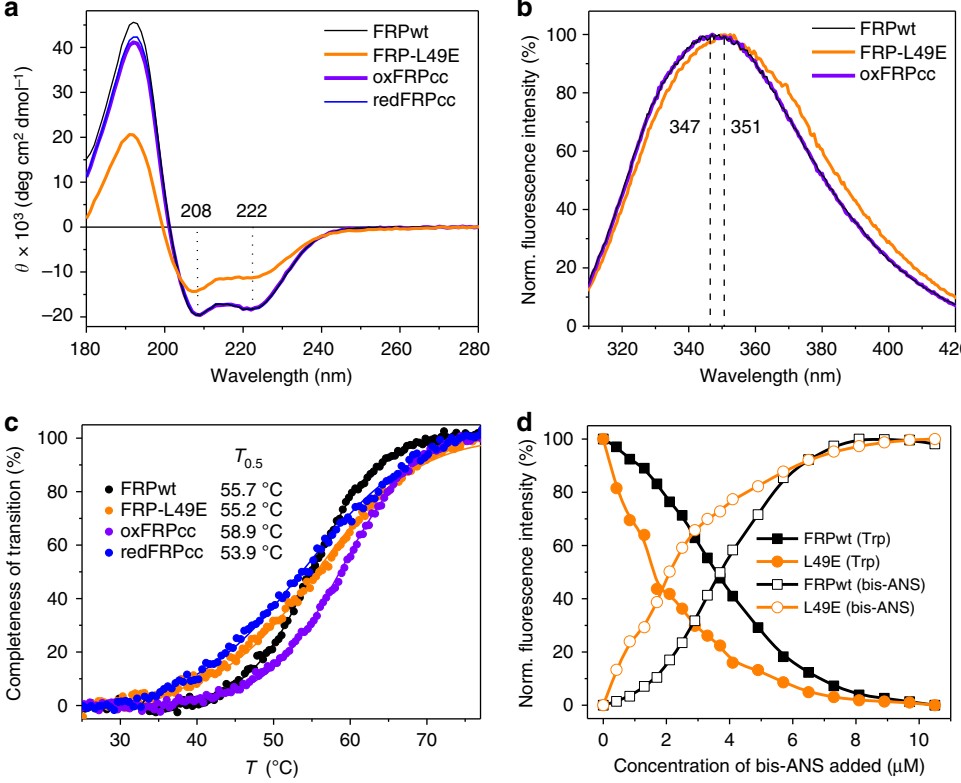

**Fig. 2** Properties of the dimeric and monomeric FRP species. **a** Far-UV CD spectra of FRPwt, FRP-L49E, oxFRPcc, and redFRPcc (at 36 μM). Positions of the peak minima are indicated in nm. **b** Intrinsic Trp fluorescence spectra for FRPwt, oxFRPcc, and FRP-L49E (at 1.6 μM). Positions of the peak maxima are indicated in nm. **c** Thermal stability of FRPwt, FRP-L49E, oxFRPcc, and redFRPcc (at 1 μM) assessed by following changes in their Trp fluorescence (excitation 297 nm; emission 382 nm) upon heating at a constant 1 °C min$^{-1}$ rate. Corresponding half-transition temperatures are indicated. **d** Titration of FRPwt (1 μM; black curves) and FRP-L49E (1 μM; orange curves) by bis-ANS (0–10.5 μM) followed by changes of either FRP Trp fluorescence (excited at 297 nm; detected at 350 nm; solid symbols) or bis-ANS fluorescence (excited at 297 nm; detected at 500 nm; open symbols) at 20 °C. See Supplementary Fig. 2 for raw spectra

**Table 1 Secondary structure elements estimated using Dichroweb[50]**

| Method | FRPwt | | | FRP-L49E | | |
|---|---|---|---|---|---|---|
| | α-Helices | β-Strands | Unstructured | α-Helices | β-Strands | Unstructured |
| CONTIN | 63.3% | 4.6% | 32.1% | 40.9% | 11.0% | 48.1% |
| SELCON3 | 65.9% | 5.1% | 29.0% | 40.0% | 12.0% | 48.0% |
| CDSSTR | 69.0% | 7.0% | 24.0% | 43.1% | 11.0% | 45.9% |

Mean residue mass 113.7 Da, calculated percentage of α-helices from FRP crystal structure (PDB ID: 4JDX) is 60.5% (75/124 residues in a dimer, unstructured N-terminal residues absent from the crystal structure are taken into account).

dimeric conformation of oxFRPcc, permitting its further utilization as FRP species unable to monomerize even at lowest protein concentrations.

**Properties of the engineered FRP mutants**. The secondary structure of the mutants was assessed by far-ultraviolet (UV) circular dichroism (CD) spectroscopy. The spectra were similar in the case of the FRPcc mutant (both under reducing and oxidizing conditions) and FRPwt and exhibited minima at 208 and 222 nm characteristic of α-helical proteins (Fig. 2a). The α-helical content predicted by different methods of the Dichroweb server (63.3–69.0%; Table 1) was close to that expected for the structural model of the His-tagged dimeric FRP construct (60.5%, or 75/124 residues). Although similar minima at 208 and 222 nm were present in the spectrum of the monomeric L49E mutant, its shape was significantly altered (Fig. 2a), reflecting reduced α-helical

content of 40.0–43.1% (Table 1). This suggests that FRP monomerization may be accompanied by local unfolding of the polypeptide chain, as previously observed for other proteins[38]. The observed ~20% reduction of the α-helical content roughly corresponds to ~25 amino acid residues within one monomer, which coincides with the length of the α-helical segment involved in dimerization (residues 33–60 in *Synechocystis* FRP). In line with this, the propensity of the latter segment to structural rearrangements is illustrated by its hinge-like role in giving two different conformations of the polypeptide chain in the crystal structure of *Synechocystis* FRP[29].

Intrinsic Trp fluorescence was used to assess the conformation of the FRP mutants since one of the two Trp residues found in *Synechocystis* FRP (Trp50) is located immediately in the subunit interface (two per dimer) and could be a good reporter of potential structural changes in its vicinity. Intrinsic Trp fluorescence spectra of FRPwt and oxFRPcc were almost

**Table 2 Structural parameters for $\Delta NTE^O$, oxFRPcc, and their complex determined by SAXS**

| | oxFRPcc dimer | $\Delta NTE^O$ monomer | 2:1 complex |
|---|---|---|---|
| Protein concentration (mg ml$^{-1}$) | 1.70 | 0.4, 3.1 (merged)[a] | 2.41 |
| **Guinier analysis** | | | |
| $I(0)$ (cm$^{-1}$) | 0.017 | 0.033 | 0.032 |
| $R_g$ (nm) | 2.91 ± 0.09 | 2.24 ± 0.05 | 3.03 ± 0.03 |
| $sR_g$ range | $0.3 < sR_g < 1.29$ | $0.28 < sR_g < 1.30$ | $0.34 < sR_g < 1.30$ |
| **$p(r)$ analysis** | | | |
| $I(0)$ (cm$^{-1}$) | 0.017 | 0.033 | 0.032 |
| $R_g$ (nm) | 3.13 ± 0.02 | 2.27 ± 0.03 | 3.15 ± 0.03 |
| $D_{max}$ (nm) | 13 | 7.4 | 13 |
| $s$ range (nm$^{-1}$) | 0.102–2.76 | 0.127–3.159 | 0.113–2.638 |
| $\chi^2$, CorMap; reciprocal space fit (GNOM estimate) | 1.02, 0.207 (0.70) | 1.21, 0.125 (0.88) | 0.99, 0.591 (0.70) |
| Kratky plot | Bell-shaped (folded) | Bell-shaped (folded) | Bell-shaped (folded) |
| **Volume, shape, and molecular weight ($M_W$) analysis** | | | |
| Porod volume (nm)$^3$ | 43.4 | 56.8 | 102.2 |
| $M_W$ calculated from amino acid sequence (kDa) | 28.2 | 34.3 | 62.4 |
| $M_W$ from Porod volume (kDa) ($M_W$ ratio)[b] | 27.2 (0.96) | 35.5 (1.03) | 63.9 (1.02) |
| $M_W$ from SAXSMoW (kDa) ($M_W$ ratio) | 29.5 (1.05) | 39.6 (1.15) | 71.0 (1.14) |
| $M_W$ from $V_c$ (kDa) ($M_W$ ratio) | 28.7 (1.02) | 34.2 (1.00) | 60.7 (0.97) |
| **GASBOR** | | | 10 calculations |
| $s$ range for fitting (nm$^{-1}$) | – | – | 0.113–2.638 |
| Number of beads (residues) | – | – | 560 |
| Symmetry, anisotropy assumptions | – | – | None |
| $\chi^2$, CorMap range (all models) | – | – | 1.01–1.08, 0.001–0.351 |
| **CORAL** | 5 calculations | 5 calculations | 20 calculations |
| $\chi^2$, CorMap range (all models) | 1.05–1.09 (0.063–0.655) | 1.21–1.27, 0.0019–0.114 | 0.99–1.03, 0.382–0.658 |
| $s$ range for fitting (nm$^{-1}$) | 0.102–3.10 | 0.127–2.88 | 0.10–3.14 |
| **CRYSOL (50 harmonics, 256 points, constant enabled)[c]** | | | |
| $s$ range for model fitting (nm$^{-1}$) | 0.102–4.44 | 0.127–4.16 | 0.11–5.00 |
| $\chi^2$, CorMap (the best model) | 1.04, 0.174 | 1.12, 0.163 | 0.99, 0.102 |
| model $R_g$ (nm) | 3.00 | 2.22 | 3.14 |
| **SASBDB Accession codes** | SASDDE9 | SASDDF9 | SASDDG9 |

GNOM, DATPOROD, DATMOW, DATVC, SASREF and CRYSOL can be found as part of the ATSAS 2.8 software package[59]
[a]Five identical samples at 0.4 mg ml$^{-1}$ were measured and their statistically similar SAXS curves were averaged to reduce noise and then merged using PRIMUS[56] with the curve obtained at 3.11 mg ml$^{-1}$ to exclude the effects of interparticle interference
[b]The experimental $M_W$ ratio relative to the calculated $M_W$ from the amino acid sequence of a dimer
[c]CRYSOL fits to the SAXS data for the whole range of scattering vectors

indistinguishable, whereas the spectrum of the L49E mutant was red-shifted by 4–5 nm (Fig. 2b). This indicated partially increased solvent exposure of Trp residues, consistent with the monomeric status of this protein.

In differential scanning fluorimetry experiments utilizing intrinsic Trp fluorescence as a readout, FRPwt underwent rather cooperative thermal unfolding with $T_{0.5} = 55.7\,°C$ (Fig. 2c). The monomeric mutant showed less cooperative unfolding, although with almost the same half-transition temperature ($55.2\,°C$) as FRPwt (Fig. 2c). The unfolding of redFRPcc was similar to that of the L49E mutant, with $T_{0.5} = 53.9\,°C$ (Fig. 2c), suggesting its compromised stability and gradual thermally-induced dissociation, rather pronounced due to the low protein concentration of the assay (1 μM). In contrast, oxFRPcc showed cooperative transition similar to that of FRPwt, but with even higher $T_{0.5}$ ($58.9\,°C$), indicating that disulfide trapped dimers resist thermal unfolding.

Since the FRP interface is stabilized by hydrophobic interactions (residues L29, L32, L33, V36, A40, I43, I46, L49, W50, L52, and L56[35]), we questioned whether FRP monomerization is associated with changes in surface hydrophobicity and compared the hydrophobic properties of FRPwt and its L49E mutant by titrating them with a fluorescent environmental probe, 4,4′-dianilino-1,1′-binaphthyl-5,5′-disulfonate (bis-ANS). Both FRP species demonstrated bis-ANS binding accompanied by a fluorescence increase and a concomitant decrease in the fluorescence of tryptophans (Supplementary Fig. 2), suggesting bis-ANS binding in their vicinity. Titration curves (Fig. 2d) showed marked differences: the monomeric FRP mutant showed sharp augmentation of bis-ANS fluorescence in the course of titration, consistent with the exposure of the hydrophobic subunit interface. FRPwt showed an appreciable lag-phase until ~2-fold molar excess of the bound bis-ANS, after which gradual rise of bis-ANS fluorescence was observed (Fig. 2d, Supplementary Fig. 2). The sigmoidal curve suggested that bis-ANS binding provoked dimer dissociation, enhancing further bis-ANS binding.

Structural properties of the oxFRPcc and L49E mutants were analyzed by SAXS. Consistent with the other data, oxFRPcc showed characteristics of the dimeric FRP (Table 2). Since its bent conformation was trapped by the engineered disulfide bridges, we fixed it and modeled the N-terminal tags using CORAL[39]. The best fitting model provided an excellent description of the data ($\chi^2 = 1.04$, CorMap 0.174; Supplementary Fig. 3a). The L49E variant showed concentration-dependent self-association, which could be expected for proteins with a pronounced exposed hydrophobicity[40]. The SAXS profiles obtained at low protein concentration were averaged and the resulting rather noisy curve was used to assess structural parameters (Supplementary Table 1). Combined with the bell-shaped Kratky plot, this analysis confirmed that, at least at low concentrations, FRP-L49E is

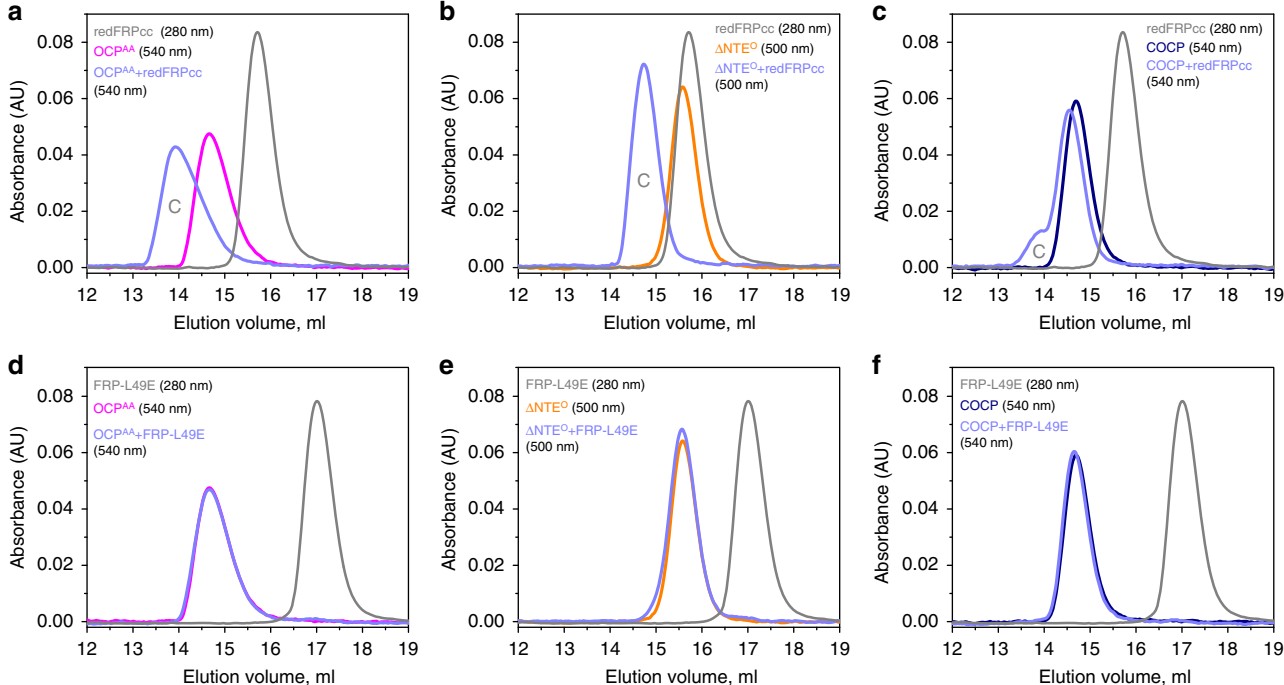

**Fig. 3** Physical interaction of the FRP mutants with various OCP forms studied by analytical SEC. Either redFRPcc (**a**, **b**, **c**) or FRP-L49E (**d**, **e**, **f**) were preincubated alone or in the presence of either OCP$^{AA}$ (**a**, **d**), ΔNTE$^{O}$ (**b**, **e**), or COCP (**c**, **f**) and then analyzed by SEC on a Superdex 200 Increase 10/300 column by following either protein-specific or carotenoid-specific absorbance (wavelengths are indicated). Distinct peaks of the complexes are marked by C. Load concentrations of FRP species, OCP$^{AA}$, ΔNTE$^{O}$, and COCP were equal to 50, 37, 6, and 8 μM, respectively

present as a rather folded monomer, however, its conformation is not equivalent to that of the crystallographic FRP subunits, as judged from the reduced α-helical content of the L49E variant (Fig. 2a). Nevertheless, the concentration dependence and the fact that its SAXS-derived parameters at 4 mg ml$^{-1}$ resembled those of the FRP dimer (Supplementary Table 1) suggest that the L49E substitution on its own does not distort the structure and leaves the residual ability to dimerize at higher protein concentrations.

**Interaction of the engineered FRP variants with OCP species.** Analytical SEC with simultaneous UV and visible detection was found particularly useful for studying the interaction between FRP and various carotenoid-bound forms of OCP[24,25,30,33]. FRP was shown to effectively bind to OCP forms with separated domains, including photoactivated OCP$^{R}$ and its constitutively active mutants, and also to OCP devoid of the NTE, as this structural element is thought to cover the FRP-binding site in OCP$^{O}$. Importantly, the ΔNTE species exists in two forms, ΔNTE$^{P}$ (purple) and ΔNTE$^{O}$ (orange), that have markedly different hydrodynamic properties, but both interact with FRP[30]. Interaction with different OCP variants represents intermediate steps of the FRP action on OCP and, given their apparent stability, may be used to analyze these steps in more detail.

In this work, comprehensively characterized FRP mutants with controlled oligomeric states allowed us to test a set of previously postulated hypotheses and examine the direct interaction of the dimeric and monomeric FRP with OCP and its derivatives. As expected, dissociable redFRPcc (Fig. 1g) showed tight interaction (i) with the constantly active mutant form carrying amino acid substitutions Y201A and W288A, OCP$^{AA}$ (Fig. 3a), (ii) with ΔNTE$^{O}$ (Fig. 3b), and (iii) with the individual CTD of *Synechocystis* OCP (amino acids 165–317), i.e., COCP, which was shown to bind carotenoids and dimerize[33] (Fig. 3c), demonstrating the binding preferences of FRPwt. Unexpectedly,

under identical conditions, no stable interactions with these OCP forms were observed for the monomeric FRP-L49E, with only traces of binding to ΔNTE$^{O}$ (Fig. 3d–f). This strongly suggests that the monomeric FRP is incompetent in OCP binding and that the previously reported FRP monomerization is unlikely to happen prior to the initial OCP recognition by FRP. More probably, monomerization takes place after the FRP dimer is already recruited by OCP. Given partial unfolding of the individual FRP monomer (vide supra), it is possible that binding of the FRP dimer to OCP is needed because FRP subunits scaffold each other by stabilizing the specific α-helical conformation competent for the interaction, whereas an already bound FRP monomer can be stabilized by contacts with OCP.

Strikingly, the monomerization-incapable oxFRPcc dimer showed very strong interaction with all the OCP forms that are known to bind FRPwt. For example, we could observe efficient interaction with OCP$^{AA}$ and COCP (Supplementary Fig. 4), as well as with both forms of ΔNTE. It not only interacted with ΔNTE$^{O}$ (vide infra), but also formed stable complexes with ΔNTE$^{P}$ having separated NTD and CTD[30] and caused its so-called oranging (Supplementary Fig. 5), i.e., tentative transition into a more compact state with connected domains and carotenoid translocation into the OCP$^{O}$-like position[24]. These completely unexpected results not only indicated that the engineered disulfide bridges did not interfere with the functionality of the protein but also that monomerization of FRP is not obligatory for its binding and action on OCP. At the same time, monomerization has been extensively evidenced earlier[24,30,32] and is well-reproduced, at least under certain conditions (see Supplementary Fig. 6a). Altogether these data imply that the dimeric interface of FRP may be disrupted during the functional OCP–FRP interaction, but is not directly involved in contacting OCP, which disproves our earlier hypothesis that the dimer interface of FRP has to be uncovered for permitting binding to OCP[24].

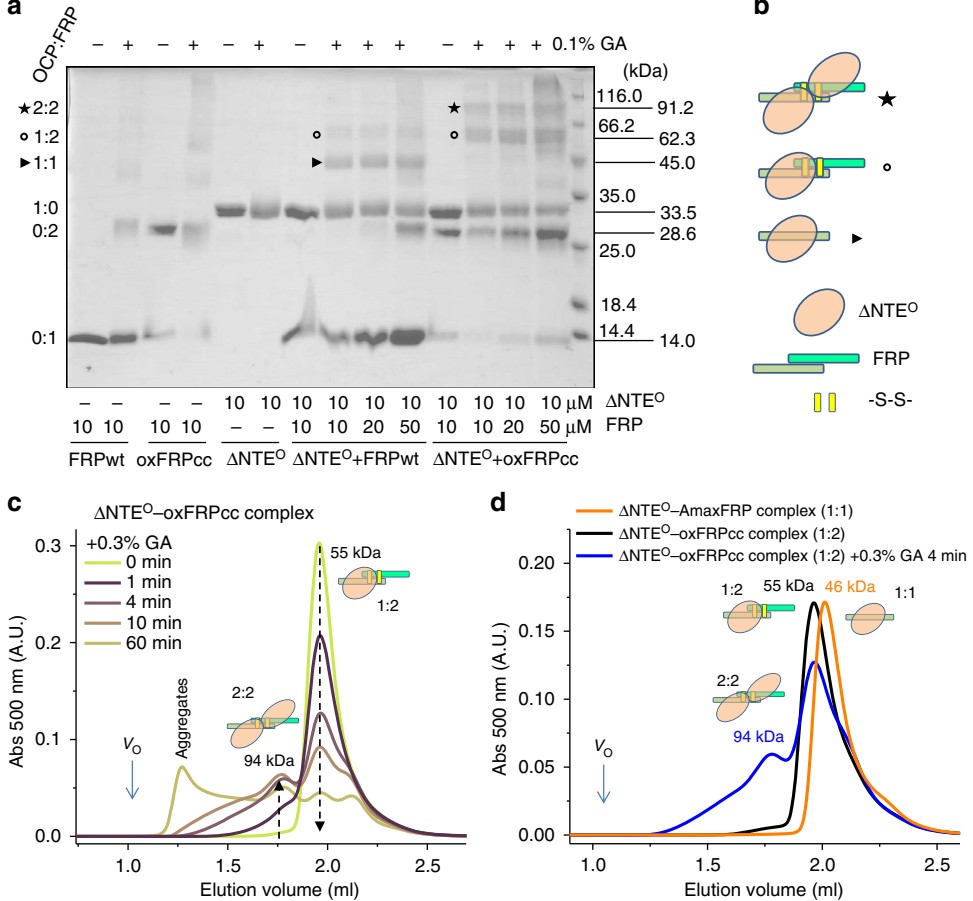

**Fig. 4** Stoichiometric analysis of the $\Delta NTE^O$–FRP interaction by GA crosslinking. **a** SDS-PAGE analysis of the results of GA crosslinking of $\Delta NTE^O$ with either FRPwt, or oxFRPcc, or of individual proteins. Protein concentrations, the presence or absence of GA, $M_w$ markers and the contents of the crosslinked samples are shown. The OCP:FRP stoichiometries corresponding to the main bands observed on the gel are given on the left, and the corresponding apparent $M_w$ are shown on the right. **b** Schematic depiction of $\Delta NTE^O$ (beige oval), the FRP dimer (tints of green) stabilized by disulfides (yellow bars), and their complexes crosslinked at different stoichiometries, relevant for **c** and **d**. Triangle, open circle, and star additionally mark the heterocomplexes with 1:1, 1:2, and 2:2 stoichiometries, respectively. **c** Kinetics of the crosslinking of the $\Delta NTE^O$ mixture with oxFRPcc by 0.3% GA (final concentration) incubated at room temperature and analyzed by SEC on a Superdex 200 Increase 5/150 column upon loading 30 μl aliquots of the reaction mixture after different incubation times. The decrease of the 1:2 complex peak and a concomitant increase of the 2:2 complex peak are marked by arrows, the void volume is indicated ($V_o$). **d** Chromatograms showing positions of the $\Delta NTE^O$–FRP complexes with different stoichiometries. SEC was followed by carotenoid-specific absorbance (500 nm). The *Arthrospira* homolog of FRP was taken because of its ability to form almost exclusively 1:1 complexes with OCP forms[25]

**Stoichiometry of the OCP–FRP interaction.** To reconcile several apparently contradictory observations, we performed GA crosslinking of the $\Delta NTE^O$ mixtures with FRPwt or oxFRPcc[30] (Fig. 4). Under the selected conditions, the individual FRP species (~14 and/or 29 kDa bands) and $\Delta NTE^O$ (~33.5 kDa band) almost did not form GA-crosslinked oligomers with $M_W > 35$ kDa that would interfere with the detection of crosslinked hetero-complexes. In line with published data, the $\Delta NTE^O$–FRPwt interaction resulted in mostly 1:1 crosslinked heterodimeric complexes (45.0 kDa) and a rather faint band corresponding to crosslinked 1:2 complexes (62.3 kDa) (Fig. 4a). The most probable intersubunit crosslinks within *Synechocystis* FRP are between residues Arg60 and Lys51 (two such pairs per homodimer). The efficiency of Arg–Lys crosslinking by GA is limited[41] and may be further lowered due to a partial masking of these residues in complexes, but also due to the spontaneous FRP monomeriza-tion. To exclude that the lack of crosslinkable residues could give the lower intensity of the 1:2 band, we took the previously characterized FRP homolog from *Anabaena*, which has four crosslinkable Lys residues in the interface, but even in this case, the efficiency of the 1:2 band crosslinking was much lower than

that of the 1:1 band (Supplementary Fig. 6b), implying that, in $\Delta NTE^O$–FRP complexes, at least partial FRP monomerization occurs.

In contrast, $\Delta NTE^O$ crosslinking with oxFRPcc resulted in 1:2 (62.3 kDa) and, strikingly, even 2:2 (91.2 kDa) complexes, whereas no 1:1 band could be detected. This strongly indicates that not only oxFRPcc remains dimeric upon OCP binding, but also that binding of two OCP molecules to one FRP dimer is principally possible (Fig. 4b). In contrast to different intensities of the 1:1 and 1:2 complex bands in the case of FRPwt, the intensities of the 1:2 and 2:2 bands in the case of oxFRPcc were similar (Fig. 4a), suggesting the potential equivalence of the binding of two OCP molecules to one FRP dimer if the latter cannot dissociate. This idea is consistent with the presence of two head domains of FRP bearing clusters of highly conserved surface residues[25]. At the same time, we could not detect such large complexes (91.2 kDa) between any OCP and FRP, but detected mainly 1:1 complexes of half of that size (~46 kDa) by SEC under equilibrium conditions (no crosslinking). This provokes the idea that consecutive binding of two OCP molecules to an FRP dimer, for some reason, is not favored and strains the conformation of

the latter provoking its dissociation, which is overcome by disulfide trapping of the FRP dimer and an irreversible process of GA crosslinking. In support of this, when we followed the kinetics of GA crosslinking of the $\Delta$NTE$^O$–oxFRPcc mixture by analytical SEC we observed gradual disappearance of the 1:2 complex and formation of higher order crosslinked species among which the distinct peak corresponding to 2:2 complexes was particularly prominent (Fig. 4c). The same situation was observed when the oxFRPcc mixture with the analog of the photoactivated OCP form, OCP$^{AA}$, was subjected to crosslinking (Supplementary Fig. 7). These experiments allowed us to compare the positions of the 1:1, 1:2, and 2:2 complexes on the chromatogram (Fig. 4d) and to conclude that 2:2 complexes are not usually detected under equilibrium conditions due to some internal tensions within OCP–FRP complexes causing their splitting into 1:1 subcomplexes. Based on this, we put forward a dissociative mechanism of the OCP–FRP interaction.

Given the low efficiency of binding of the FRP monomer (Fig. 3d–f) and the ineffective formation of 2:2 complexes under equilibrium conditions (no crosslinking), binding of the FRP dimer to OCP should be the primary stage that could be followed by SEC at a low OCP concentration and varying concentrations of oxFRPcc (Fig. 5a). Under these conditions, we found almost identical binding curves for oxFRPcc and dissociable FRPwt with a submicromolar apparent $K_d$ (Fig. 5b). We cannot exclude that the primary binding induces some conformational change that weakens the FRP interface on its own; however, consecutive binding of two OCP molecules is expected to play an active role in disrupting FRP dimers. Biophysical modeling of this situation in different concentration regimes is described in the Supplementary Note 1.

**Topology of the $\Delta$NTE$^O$–oxFRPcc complexes**. Despite the acquired ability to obtain highly pure and stable complexes with controlled stoichiometry, extensive crystallization screening of various OCP–FRP complexes (>5000 conditions overall) failed so far. This could be related to the dynamic nature of the desired complexes, existing in an equilibrium between the states in which either OCP represents an intermediate of its photocycle or FRP is detached from OCP, since its functional activity (alignment of the CTD and NTD) is already complete (see Supplementary Fig. 8). These factors forced us to characterize the OCP–FRP interaction using SAXS and complementary techniques.

To avoid the necessity of dealing with the high conformational flexibility of photoactivated OCP analogs with separated domains, we focused on the analysis of the FRP complex with the compact $\Delta$NTE$^O$ having the exposed FRP binding site on the CTD[30], which represents an intermediate of the OCP compaction process associated with the alignment of OCP domains, immediately preceding FRP detachment and termination of its action cycle.

First, we verified that individual $\Delta$NTE$^O$ adopts a compact conformation equivalent to that in OCP$^O$. The SAXS data for relatively low protein concentrations revealed structural properties in solution expected from the compact OCP$^O$ monomer (Table 2), supported also by the $p(r)$ distribution function (Fig. 5c). Consistently, a crystallographic model of OCP$^O$ devoid of the NTE provided an excellent fit to the data ($\chi^2 = 1.12$, CorMap 0.163, see Supplementary Fig. 3c,d). The disulfide-trapped oxFRPcc dimer was characterized above (Supplementary Fig. 3).

SAXS analysis of the $\Delta$NTE$^O$–oxFRPcc complex concentrated to 2.41 mg ml$^{-1}$ (~40 $\mu$M), where the complete binding occupancy was expected (Fig. 5a), suggested particles with a size expected for the 1:2 complex ($M_W$ Porod = 63.9 kDa; calculated $M_W$ = 62.4 kDa, Table 2), allowing construction of its

low-resolution structural model. Complex formation was nicely reflected in the $p(r)$ distribution function characterized by a combination of features of the elongated FRP dimer and the globular OCP monomer (Fig. 5c). The FRP dimer was fixed due to the presence of interfacial disulfides, $\Delta$NTE$^O$ was taken as the N-terminally truncated part of the compact OCP$^O$, and their relative position as well as short N-terminal tags on both FRP and OCP, were modeled using CORAL[39], without imposing any contact restraints. The resulting models provided excellent fits to the SAXS data ($\chi^2 = 0.99$–1.03 among 20 models), but differed by the relative orientation of the FRP dimer and OCP. The majority of the models had FRP contacting OCP-NTD only and were discarded. Among the models with FRP contacting OCP-CTD, which is thought to harbor the main FRP-binding site[24,29,30,33,34], one had the FRP dimer lying along OCP where the concave side of FRP (involving highly conserved residues such as R60) was simultaneously contacting the OCP-NTD (Fig. 5d).

Remarkably, in this model, which describes the SAXS data exceptionally well (Fig. 5e), one of the FRP head domains contacts the NTE binding site involving the key F299 residue on the β-sheet surface of the CTD[42], whereas the second head domain and the FRP dimeric interface are not engaged (Fig. 5d). In excellent agreement with the results of GA crosslinking, this leaves the possibility of binding two OCP molecules using the two valences located symmetrically on head domains of FRP; however, most notably, an apparent clash between parts of the simultaneously bound OCP molecules takes place (Fig. 5f). It is reasonable to suggest that this steric hindrance may create internal tension in the 2:2 complex, causing its splitting into 1:1 subcomplexes in the case of FRPwt. In the oxFRPcc case, this could explain the low efficiency of binding of the second OCP, unless this stoichiometry is fixed by chemical crosslinking (Fig. 4).

Importantly, our model is consistent with the data of mutational studies and crosslinking mass-spectrometry[29,34,42] (Supplementary Fig. 9). In particular, F299 of OCP and F76 and K102 of FRP belong to the OCP–FRP binding region predicted by our model (Figs. 5c and 6a) and both F76 and K102 form highly conserved clusters on both head domains of FRP (Fig. 6a), emphasizing the importance of these residues and indirectly supporting the discussed topology of the OCP–FRP complexes. Such a scenario is also supported by the complementary distribution of electrostatic surface potentials on the interface of interacting proteins, suggesting that the FRP dimer with an extended negatively charged surface between the positively charged head domains serves as a scaffold for the re-assembly of the CTD and NTD exhibiting complementary clusters of opposite charge (Fig. 6b). Unfortunately, the inherently low resolution of the SAXS-derived model does not allow us to consider any drastic conformational changes within the interacting partners, for example, those involving the recently proposed unfolding of the head domains[34], that, in principle, is very likely and compatible with our model. The dynamic nature of the OCP–FRP interaction makes it an extremely complicated regulatory system warranting further structural studies.

To covalently trap the OCP–FRP heterocomplex, we engineered mutant forms with the key interface residues, K102, F76 of FRP[34] and F299 of OCP[42] (Fig. 6a), replaced by cysteines. OCP-F299C was isolated as a stable photoactive protein being capable to undergo R–O conversion by FRPwt (~20-fold acceleration; Fig. 6c). The F76C and K102C mutants of FRP accelerated the R–O conversion of OCP-F299C with efficiency decreasing in the sequence FRPwt >> FRP–K102C > FRP–F76C (Fig. 6c), underlining the importance of these residues for the OCP–FRP interaction. The preservation of this interaction,

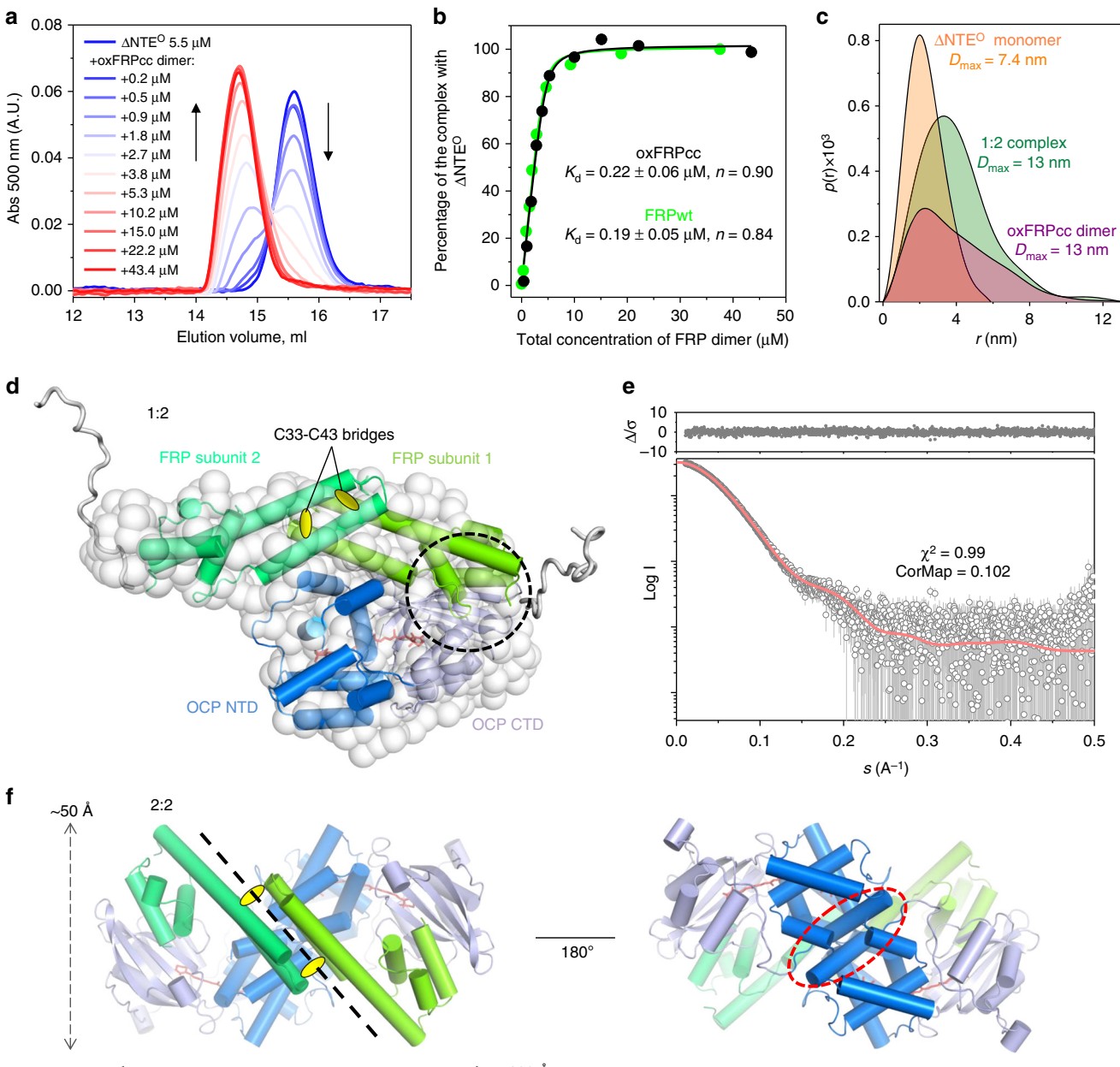

**Fig. 5** Analysis of the $\Delta NTE^O$ interaction with oxFRPcc. **a** A fixed concentration of $\Delta NTE^O$ was titrated by increasing amounts of oxFRPcc (indicated in μM per dimer); the samples (100 μl) were analyzed using a Superdex 200 Increase 10/300 column in the absence of reducing agents. Arrows indicate the direction of titration. **b** The binding curve obtained upon quantification of the amplitude of the $\Delta NTE^O$–oxFRPcc peak presented in **a**, in comparison with the curve for FRPwt (identical conditions). **c** Pairwise distance distribution functions for $\Delta NTE^O$, oxFRPcc dimer, and their complex obtained using GNOM. **d** One of the possible conformations of the $\Delta NTE^O$–oxFRPcc complex (1:2) consistent with the SAXS data and complementary information, shown as the CORAL-derived atomistic model overlaid with the best fitting GASBOR-derived ab initio bead model. Dashed circle in **d** marks the tentative FRP binding site located on the β-sheet of the OCP-CTD, normally occupied by NTE in $OCP^O$. **e** The fit of the CORAL model to the SAXS data with the associated residuals ($\Delta/\sigma$). **f** Hypothetical 2:2 binding on top of the 1:2 complex suggested by crosslinking experiments. Although two tentative OCP-binding sites on the head domains of FRP may coexist, the 2:2 binding leads to a clash between OCP molecules (marked by a red dashed circle). In the dissociable FRPwt, such a binding may provoke FRP monomerization and formation of the 1:1 heterocomplexes to relieve tension caused by the clashing OCP molecules. In oxFRPcc, this is not possible because of the covalent interface stabilization by disulfides

though diminished, left the possibility of oxidative disulfide crosslinking if the corresponding residues are proximal in native complexes (Fig. 6d). Dialysis in the presence of GSH/GSSG indeed produced a 46 kDa band expected for 1:1 complex on SDS-PAGE under non-reducing conditions in the F299C–K102C combination, whereas no such band could be detected in the F299C–F76C combination, or if the sample was reduced by βME (Fig. 6e). This band was absent in individual samples, and, therefore, could only correspond to the heterocomplex trapped by the F299C–K102C bond (Fig. 6e). This directly confirms the spatial proximity of the F299 and K102 residues in the OCP–FRP complexes and strongly supports the proposed topology (Fig. 5d, f). We cannot exclude that F299/F76 residues are also neighboring in heterocomplexes and contribute to the interaction (Fig. 6d), because their mutation significantly compromises it, making the probability of Cys–Cys crosslinking lower than in the F299–K102

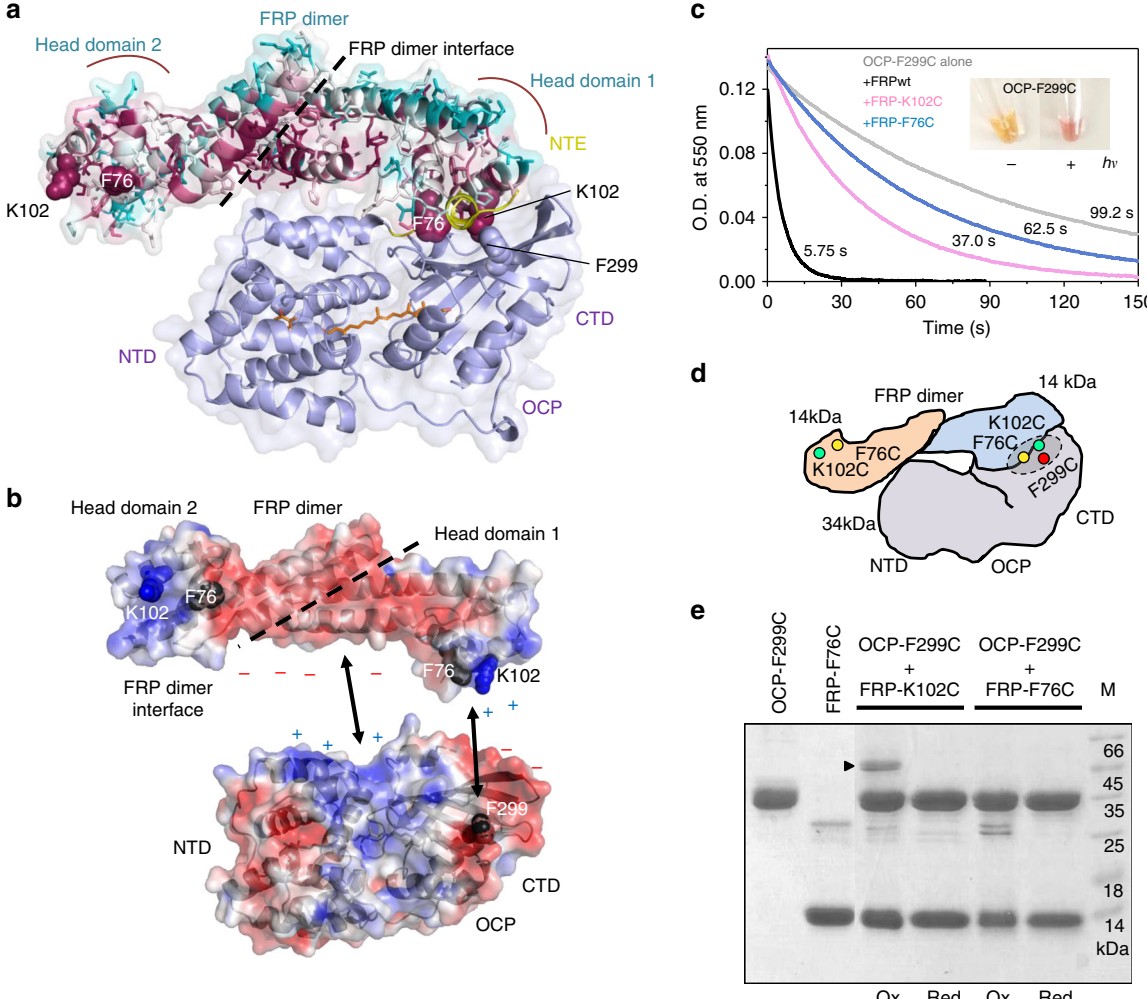

**Fig. 6** Validation of the proposed topology of the OCP–FRP complexes. **a** The SAXS-derived structural model of the 1:2 ΔNTE$^O$–FRP complex with FRP residues colored by a gradient from conserved (purple) to variable (cyan) using Consurf[65]. OCP is shown in light-violet with the carotenoid in orange. Note high conservation on the concave side of the FRP dimer and that (i) binding of the first head domain of FRP occurs on the OCP-CTD in place of the NTE (shown in yellow), (ii) presumable contact area includes F299 of OCP and K102 and F76 of FRP, whereas (iii) the second head domain of FRP is open for the interaction with another OCP molecule and (iv) the dimer interface of FRP is not directly involved in OCP binding. **b** Distribution of the regions with positive ($+3\,k\,T\,e^{-1}$; blue) and negative ($-3\,k\,T\,e^{-1}$; red) electrostatic potentials on surface of FRP and OCP suggesting extended multisite binding, in agreement with the scaffolding role of FRP. **c** Functional interaction of Cys mutants of OCP and FRP assessed by the ability of FRP variants to accelerate the R–O conversion of the photoactivated OCP-F299C at 25 °C. Insert shows the color of the OCP-F299C sample in the dark and under actinic light. **d** Schematic picture of the 1:2 complex with the positions chosen for Cys mutagenesis and disulfide trapping. The dashed circle indicates the tentative OCP–FRP interface. **e** The ability of Cys mutants to form disulfide crosslinked heterocomplexes upon mild oxidation by GSH/GSSG of the OCP-F299C mixtures with either FRP-K102C or FRP-F76C mutants. $M_w$ markers (M) are indicated in kDa. Ox and Red designate the absence or presence of βME in the sample buffer. Arrowhead marks the 46 kDa band corresponding to the OCP–FRP complex fixed by disulfide bond and disappearing upon reduction

case. But, this contrast further supports the notion that F299 and K102 belong to the OCP–FRP interface.

**The effect of FRP species on the R–O conversion of OCP.** The role of the oligomeric state of FRP on its functional activity was analyzed by the ability of FRPwt and mutants thereof to accelerate the R–O conversion of wild-type OCP. Under conditions used, OCP$^R$ slowly converts to OCP$^O$, which can be followed by the decrease of absorbance at 550 nm (Fig. 7a). Consistent with its physiological role, FRPwt accelerates the R–O transition by providing a scaffold which OCP needs to explore a smaller number of configurations regarding the relative position of its domains to restore the basal compact conformation[15,24]. In line with its inefficient binding with OCP forms, the monomeric FRP-L49E mutant displayed only marginal acceleration of the R–O

transition, whereas oxFRPcc showed intermediate activity (Fig. 7a). By titrating OCP with increasing amounts of FRP species and following the steady-state level of the R–O conversion under continuous illumination we could analyze their effectiveness in more detail (Fig. 7b, c). These experiments showed that the decrease of maximally achievable concentration of OCP$^R$ with separated domains reaches saturation at a FRP/OCP ratio >2 and increases in the sequence FRPwt > oxFRPcc >> L49E (Fig. 7b). FRPwt is the most efficient facilitator with minimal half-saturation stoichiometry (~0.35 FRP monomer/OCP), rather than oxFRPcc (~0.80) (Fig. 7b, insert), in complete agreement with the proposed dissociative mechanism. The same pattern was observed from comparison of the R–O conversion rates (FRPwt > oxFRPcc >> L49E; Fig. 7c). Although we cannot rule out that introduction of Cys–Cys bridges somehow reduced the

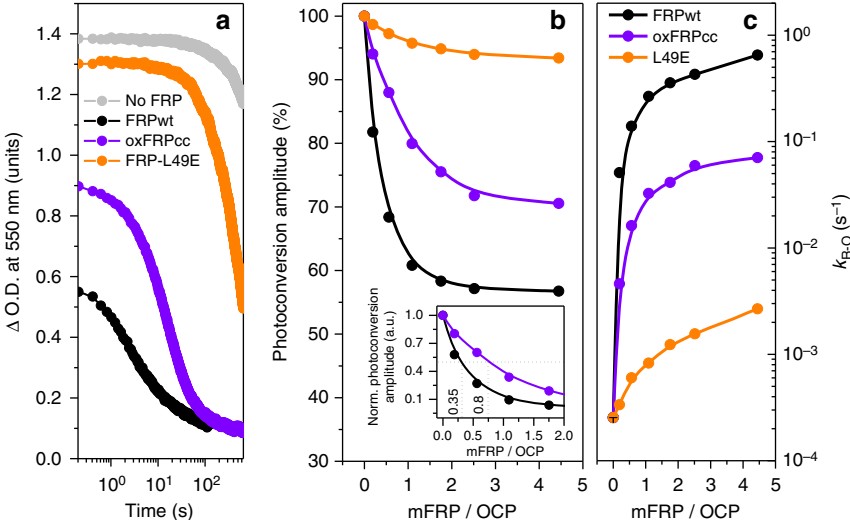

**Fig. 7** Functional characterization of FRP variants with predefined oligomeric state. **a** Characteristic time-courses of $OCP^R$-$OCP^O$ relaxation in the absence or presence of FRP species [a fixed ratio of ~1.7 FRP per OCP; monomeric FRP concentration (mFRP) was chosen] followed by changes of optical density (O.D.) at 550 nm after the actinic light is turned off. Maximal O.D. changes at 550 nm which could be obtained in the presence of FRP species under constant illumination by the actinic light (**b**) – normalized to such values in the absence of FRP species, and, thus, representing the maximal concentration of $OCP^R$ normalized to values between 0 and 1 for dimeric FRP variants to show at which FRP/OCP ratio half-saturation occurs (insert). **c** Corresponding R-O conversion rates in the presence of different concentrations of FRP species. All experiments were conducted at 10 °C to reduce the rate of $OCP^R$-$OCP^O$ conversion, which is otherwise extremely high in the presence of FRPwt

flexibility of the FRP dimer and by this means contributed to its lower efficiency, our data support the advantageous role of the FRP monomerization.

## Discussion

By using an integrative approach and uniquely engineered FRP and OCP mutants, this study provides important mechanistic insights and allows to propose a dissociative mechanism of FRP function (numbers of reaction steps below refer to those in Fig. 8). The photoactivation of OCP leads to detachment of the NTE, separation of OCP domains, and the translocation of carotenoid to form $OCP^R$ (1), which slowly relaxes to the basal $OCP^O$ form in the dark. The NTE detachment enables binding of the FRP dimer at the NTE-binding surface on the CTD via the head domain of FRP (2), as directly demonstrated here by disulfide trapping using OCP-F299C and FRP-K102C mutants, whereas monomeric FRP cannot bind efficiently, probably because it lacks the proper α-helical conformation. The 1 OCP to 2 FRP binding stoichiometry provides a scaffold for the separated OCP domains facilitating their mutual approach, which is observed as oranging of the otherwise red-purple $OCP^R$ or its analogs, but allows for spontaneous FRP monomerization (1:1 complex). The dimeric interface of FRP is not involved in contacting OCP and may weaken as a result of binding per se or due to conformational rearrangements within the complex. However, transient pseudo-symmetric binding of the second OCP molecule to the 1:2 complex (2:2 complex) using the second head domain of FRP (3a) leads to a tentative clash between the two OCP molecules (3b), which provokes splitting of the 2:2 complex into 1:1 sub-complexes (4). Upon either 1:1 or 1:2 complex formation, the FRP-assisted recombination of the OCP domains enables carotenoid back-translocation (5). Reconnection of the OCP domains on the FRP scaffold allows the NTE to facilitate detachment of the bound FRP and restore the basal OCP conformation (6) ready for further photoactivation. As demonstrated by comparison of the wild-type, dissociable, and the constantly dimeric FRP variant, monomerization is not mandatory for

functional activity of FRP, but may significantly improve its efficiency, especially at elevated concentrations of $OCP^R$.

The FRP–FRP and FRP–OCP molecular interfaces and the topology of the heterocomplexes identified here are not only key for fundamental understanding of the regulatory processes conferring high light tolerance in cyanobacteria but may also inspire future developments of innovative optogenetic systems transducing light signals into protein–protein interactions, alternative to those based on bacterial and plant phytochromes, light-oxygen-voltage (LOV) domain proteins, and blue light using FAD (BLUF) domain proteins[43–48].

## Methods

**Proteins**. The His6-tagged wild-type *Synechocystis* FRP (residues 1–109; unclea-vable tag) was cloned into pQE81L vector by BamHI/HindIII endonuclease restriction sites[24,32] and used as the template to obtain the putatively monomeric L49E mutant or the FRPcc (L33C/I43C) mutant by site-directed mutagenesis using the megaprimer method;[49] for which the L49E-forward or the L33C/I43C reverse and the corresponding pQE (Qiagen) vector-specific (T5 forward and pQE reverse) primers were used (see Supplementary Table 2). The PCR products were gel-purified and cloned into a modified pQE81L plasmid (ampicillin resistance) by BamHI/HindIII endonuclease restriction sites. The identity of the constructs and the presence of mutations were verified by DNA sequencing (Evrogen, Moscow, Russia). The obtained plasmids were used to transform chemically competent cells of *Escherichia coli* M15[pREP4] strain. Proteins were expressed using induction by 1 mM isopropyl-β-thiogalactoside (IPTG) in the presence of kanamycin and ampicillin. Alternatively, the FRPcc mutant was expressed in T7 SHuffle cells (New England Biolabs, NEB) in the presence of ampicillin only, to exploit oxidized intracellular conditions and disulfide isomerase DsbC for improving the yield of the disulfide crosslinked dimeric protein.

The FRP-F76C and K102C mutants were generated using F76C forward and F76C reverse or K102C forward and K102C reverse primers (see Supplementary Table 2), respectively, and the Q5-site-directed-mutagenesis kit from NEB. The mutated FRP constructs were cloned into a pQE81L-derived vector (termed pQE81M) by using BamHI and NotI endonuclease restriction sites. The pQE81M plasmid includes a human rhinovirus 3C protease site (LEVLFQ/GP) for cleaving off the His6-tag (resultant amino acid sequence at the N-terminus after cleavage of His6-tag: GPDPATM**1**LQ...). The cDNA of the OCP-F299C mutant was also generated by using the Q5-site-directed-mutagenesis kit of NEB using F299C forward and F299C reverse primers (see Supplementary Table 2). The mutated cDNA was cloned into a pRSFDuet-derived vector (termed pRSFDuetM) by using BamHI and NotI restrictions sites and verified by DNA sequencing (Eurofins MWG Operon, Ebersberg, Germany). The pRSFDuetM-vector also encodes a His6-

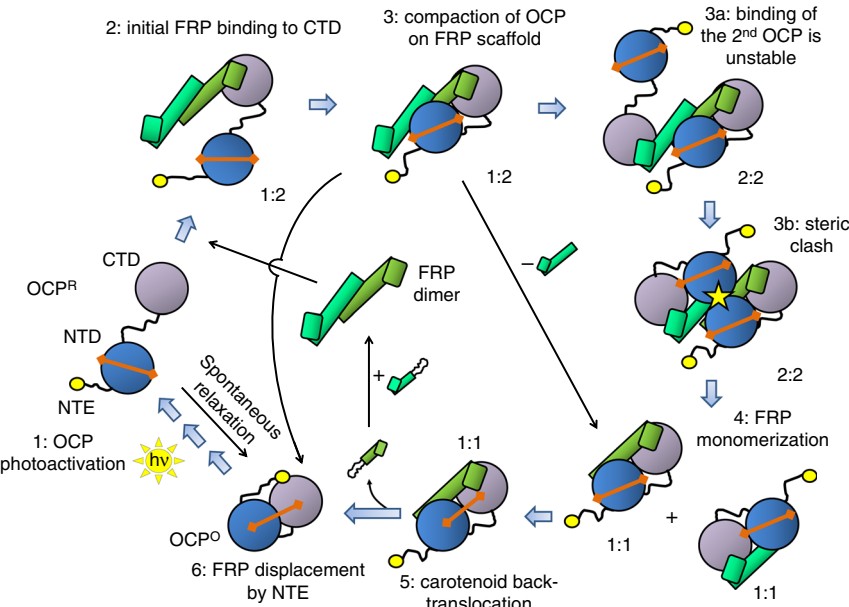

**Fig. 8** Proposed mechanism of the FRP scaffold terminating OCP-mediated photoprotection in cyanobacteria. Stages of the process are numbered from 1 to 6 and described in the text. The proteins are color-coded as in Fig. 5d, the carotenoid is shown as an orange dumbbell. OCP photoactivation is depicted by sun symbol. Yellow circle designates NTE. Individual FRP monomers are shown partially unfolded. Stoichiometry of the heterocomplexes formed between OCP and FRP are indicated. The yellow star designates the tentative clash between two bound OCP molecules destabilizing OCP–FRP complexes with 2:2 stoichiometry

tag which is cleavable by 3C protease yielding the N-terminal amino acid sequence of OCP: GPDPATM[1]PFT…. *Arthrospira maxima* and *Anabaena variabilis* FRP homologs were obtained by artificial gene synthesis (GeneArt, Regensburg, Germany), subcloned into pQE81M plasmids via BamHI/NotI restriction sites and verified by DNA sequencing (Eurofins MWG Operon, Ebersberg, Germany).

Holoforms of *Synechocystis* wild-type OCP, OCP-F299C, ΔNTE, COCP, and OCP[AA] were expressed in echinenone (ECN) and canthaxanthin (CAN)-producing *E. coli* cells[24,25,30,33] using induction with 0.5 mM IPTG during 3 d at 25 °C. All His[6]-tagged proteins were purified by immobilized metal-affinity chromatography and SEC to electrophoretic homogeneity and stored at 4 °C in the presence of 3 mM sodium azide (OCP) or frozen at −80 °C (FRP). Protein concentrations were determined spectrophotometrically using calculated protein-specific molar extinction coefficients. The obtained holoprotein preparations exhibited visible-to-UV absorption ratios of 1.6–1.8 (in case of COCP—2.5), indicating a low content of their apoforms. In the case of OCP[AA], however, low apoform content could not be achieved and the visible-to-UV absorption ratio was ~0.5. Each protein was expressed and purified at least twice and batches demonstrated similar properties.

**Oxidation of the Cys mutants**. FRPcc was first expressed in *E.coli* T7 SHuffle express cells (NEB) and purified in the absence of reducing agents, which on its own led to incomplete Cys–Cys oxidation. To optimize FRPcc oxidation, various conditions were examined. 100 µl of FRPcc samples (52 µM per monomer) were dialyzed against 100 ml of 50 mM Tris-HCl buffer (pH 7.6) without additives (control) or in the presence of 10 µM ZnSO₄, 1 mM H₂O₂, or the GSH/GSSG pair (1 mM each) for 2 d at 4 °C. The efficiency of crosslinking was assessed by SDS-PAGE in the absence or presence of 20 mM βME. Dialysis against 1 mM GSH/GSSG was found to be most efficient lacking adverse effects; the best results (~95% crosslinking) were achieved upon 8 d oxidative dialysis at 4 °C in the presence of 0.01 mM phenylmethylsulfonyl fluoride (PMSF) and 3 mM sodium azide. The oxidized FRPcc in its dimeric state was stable to reduction, requiring high concentration of dithiothreitol (DTT) or β-mercaptoethanol (βME) and significant time to completely disassemble the dimer, indicating that the formed disulfide bridges are not easily solvent-accessible, in line with their rather buried position in the protein structure.

To assess the possibility of further crosslinking, GA was added to either FRPwt or oxFRPcc at a final concentration of 0.1% for 15 min at room temperature and the results were analyzed by 15% SDS-PAGE in the absence of reducing agents.

Oxidation of the mixtures of the OCP–F299C apoform with either FRP–F76C or FRP–K102C was achieved in the dark by dialysis at 4 °C against GSH/GSSG as described above for FRPcc. Alternatively, 30–60 min oxidation by 10–50 mM H₂O₂ was tested, both yielding crosslinking almost exclusively in the F299C–K102C combination. The efficiency of crosslinking was assessed by SDS-PAGE in the absence or presence of 20 mM βME.

**CD spectroscopy**. Far-UV CD spectroscopy was used to assess the secondary structure of the different FRP forms. Protein samples (0.5 mg ml⁻¹, or 36 µM per monomer) in 30 mM phosphate buffer (pH 7.2) were measured at 25 °C using a Chirascan dichroism spectrophotometer (Applied Photophysics) equipped with a thermostated cell and a 0.2 mm cuvette. Measurements were done in the range 180–280 nm in 1 nm steps and a slit width of 1 nm. Three spectra for each sample were recorded, baseline corrected and averaged. The secondary structure elements were calculated by decomposition of the CD spectra for FRPwt and FRP–L49E using Dichroweb[50]. Results are summarized in Table 1.

**Steady-state fluorescence measurements**. Intrinsic Trp fluorescence spectra were recorded on a Cary Eclipse spectrofluorometer (Varian) equipped with a thermostatted multicell holder. Protein samples (1 µM per monomer) were prepared on 0.22 µm filtered buffer F (20 mM Hepes-NaOH, pH 7.5, 100 mM NaCl). Fluorescence was excited at 297 nm and recorded in the range 305–450 nm (slits width 5 nm, detector voltage 700 V) at 20 °C. Subsequently, the spectra were buffer-subtracted and normalized.

To assess the hydrophobic properties of FRP species, 1 µM protein samples in buffer F were titrated by increasing amounts of aqueous stock solutions of bis-ANS (200 µM) so that the final concentration of the fluorescence probe was in the range of 0–10.5 µM. Fluorescence was recorded after each 0.5–1 µl addition of the bis-ANS probe in two spectral channels simultaneously (Trp and bis-ANS; excitation at 297 nm, emission in the range 305–590 nm) or only in the bis-ANS channel (excitation at 385 nm, emission in the range 415–590 nm). Bis-ANS concentration was determined using a molar extinction coefficient of 16,790 M⁻¹ cm⁻¹ at 385 nm[51].

**Thermal stability of FRP species**. To assess thermally-induced changes in FRP oligomeric state, we analyzed changes in their intrinsic Trp fluorescence (excitation at 297 nm; emission at 382 nm; slit width 5 nm, detector voltage 700 V) upon heating of 1 µM protein samples prepared in buffer F at a constant rate of 1 °C min⁻¹ on a Cary Eclipse spectrofluorometer (Varian) equipped with a multicell holder and a Peltier temperature controller. The raw temperature dependencies, showing a single thermal transition, were transformed into dependences of completeness of transition on temperature[52,53] by linear approximation of the regions before and after the transition and representation of the data as percentage of the transition from the folded to the unfolded state. From these transformations, half-transition temperatures ($T_{0.5}$) were directly determined. The experiment was repeated in triplicate and the most typical results are presented.

**Native PAGE**. Protein samples containing FRP (1 mg ml⁻¹) were analyzed by electrophoresis in the glycine-Tris gel system under non-denaturing conditions[24,54]. Electrode buffers and gels contained uniform concentration of

glycine (80 mM) titrated by Tris to pH values of 8.6. The gels were run at 350 V and stained by Coomassie brilliant blue.

**Analytical SEC.** Oligomeric state of FRP species and their interaction with various OCP forms were analyzed by SEC on either Superdex 200 Increase 10/300 or Superdex 200 Increase 5/150 columns (both GE Healthcare) operated using a ProStar 325 chromatographic system (Varian) with simultaneous UV/vis detection.

In the first case, protein samples containing FRP species at different protein concentrations (1–40 μM per monomer) were pre-incubated for at least 20 min at room temperature, and then separated by the column equilibrated with the SEC buffer (20 mM Tris-HCl, pH 7.6, 150 mM NaCl, 0.1 mM EDTA, and 3 mM βME) and calibrated using the bovine serum albumin (BSA) monomer (66 kDa), the BSA dimer (132 kDa), the BSA trimer (198 kDa), and the α-lactalbumin monomer (15 kDa). The samples containing the pre-oxidized FRPcc mutant were analyzed by SEC in the absence of reducing agents; however, additional tests revealed that oxFRPcc dimers withstood even very long incubations in the presence of reducing agents without disassembly. The elution profiles were followed by absorption at 280 nm.

In the second case, protein samples containing individual FRP (or its mutants), ΔNTE, COCP, $OCP^{AA}$, or the FRP/OCP mixtures at different protein concentrations were used to study direct protein–protein interactions[24,25,30]. Protein concentrations and load volumes are specified in each particular case. The elution profiles were followed by simultaneously recording 280 nm and carotenoid-specific absorbance (wavelengths are specified in the figures). Typical results obtained in at least three independent experiments are presented.

To assess binding parameters, $ΔNTE^O$ was titrated by either FRPwt or oxFRPcc, and the amplitude of the peak of the complexes was used to plot binding curves against the total concentration of the FRP dimer. The approximation was done using the quadratic equation to estimate the apparent dissociation constants[24,30]. The experiments were repeated three times and the most typical results are presented.

**Chemical crosslinking by GA.** Protein samples containing either FRP species, $ΔNTE^O$, or their mixtures (total volume 40 μl) were pre-incubated in 20 mM Hepes-NaOH buffer, pH 7.5, 150 mM NaCl, 0.1 mM EDTA for 15 min at room temperature. Then, freshly prepared GA was added up to a final concentration of 0.1% for 25 min at room temperature. The results of the crosslinking were analyzed by 15% SDS-PAGE. Samples containing oxFRPcc were analyzed in the absence of reducing agents to preserve the disulfide crosslinked FRP dimers. The experiment was repeated two times with the qualitatively similar results.

In addition, the kinetics of GA crosslinking was analyzed by incubating either $OCP^{AA}$ or $ΔNTE^O$ mixtures with oxFRPcc in the presence of 0.1–0.3% GA (final concentration) at room temperature, and by analyzing 30 μl aliquots of the reaction mixture by SEC on a Superdex 200 Increase 5/150 column at a 0.45 ml min$^{-1}$ flow rate following 280 nm and carotenoid-specific absorbance simultaneously. The maxima of the peaks were used to assess $M_W$ values using column calibration as described above.

**SAXS analyses.** SAXS data ($I(s)$ versus $s$, where $s = 4πsinθ/λ$, $2θ$ is the scattering angle and $λ = 1.24$ Å) from samples of the engineered mutants of *Synechocystis* FRP, $ΔNTE^O$ and its complex with oxFRPcc were measured at 10 °C at the EMBL P12 beamline (PETRA III, DESY Hamburg, Germany)[55] using a Pilatus 2 M detector and a batch mode in a common matched buffer SEC containing 3% v/v glycerol, and 2 mM DTT instead of βME (for FRP–L49E and $ΔNTE^O$), or not containing reducing agents at all (for oxFRPcc and $ΔNTE^O$–oxFRPcc). Collecting series of frames (1 s exposure time, collected as 20 × 50 ms frames) for each sample revealed no radiation damage. The SAXS data collected at sample concentrations of 0.8–4 mg ml$^{-1}$ (FRP–L49E) or 0.55–5.1 mg ml$^{-1}$ (oxFRPcc) showed concentration dependence above 2 mg ml$^{-1}$ and, therefore, the data obtained at lower concentrations (1.7 mg ml$^{-1}$ for FRPcc and 1.2 mg ml$^{-1}$ for FRP–L49E) were used for further analysis of the dimeric and monomeric forms of FRP, respectively. The SAXS data for FRP–L49E at the highest concentration (4 mg ml$^{-1}$) were also used to extract structural parameters (Supplementary Table 1). The SAXS data for $ΔNTE^O$ were collected at 0.4–3.11 mg ml$^{-1}$. To minimize the effect of inter-particle interference, five identical samples at 0.4 mg ml$^{-1}$ were used to get the low concentration curve, which was then merged with the curve collected at 3.11 mg ml$^{-1}$ using merge function in PRIMUS[56]. The SAXS data for the $ΔNTE^O$–oxFRPcc complex were collected at 1.2–2.41 mg ml$^{-1}$ and those at highest concentration were used for further analysis of the structural parameters and modeling.

Data reduction, radial averaging and statistical analysis (e.g., to detect radiation damage, or scaling issues between frames) were performed using the SASFLOW pipeline[57]. Statistically similar SAXS profiles (based on CorMap[58]) were averaged and the buffer scattering subtracted to produce $I(s)$ versus $s$ scattering profiles. ATSAS 2.8[59] was employed for data analysis and modeling. PRIMUS[56] was used to perform Guinier analysis, from which the radius of gyration, $R_g$, and extrapolated zero-angle scattering, $I(0)$, were determined (Table 2). The probable frequency of real-space distances, or $p(r)$ distributions, were calculated using GNOM[60] providing additional $R_g$ and $I(0)$ estimates and the maximum particle dimension,

$D_{max}$. The Porod volume was used to assess an $M_W$ value using an empirical constant equal to 1.6[39]. Independent $M_W$ estimates were also obtained using SAXSMoW[61] and volume-of-correlation, $V_c$[62], approaches. Results are presented in Table 2 and Supplementary Table 1.

For FRPcc dimer modeling, the engineered disulfide bridges were artificially introduced in PyMOL. To account for the 22 N-terminal residues present in the construct, but absent from the crystallographic structure (PDB ID: 4JDX, chains B and D), we used modeling in CORAL[39] that minimized the discrepancy between the model-derived SAXS profile and the experimental SAXS data collected for the oxFRPcc dimer. Modeled scattering intensities were calculated using CRYSOL[63].

The structural model of $ΔNTE^O$ was obtained based on the OCP$^O$ monomer (PDB ID: 4XB5), which was first truncated to remove NTE (residues 1–20). Then, 13 N-terminal residues present in the construct were modeled by CORAL[39].

To model the structure of the $ΔNTE^O$–oxFRPcc complex (1:2), the proteins were supplemented with N-terminal residues absent from their atomistic structures (22 in each FRP chain and 13 in ΔNTE) and their relative position was systematically changed using CORAL[39] to minimize the discrepancy between the calculated scattering profile and the experimental data. The FRPcc dimer was fixed, whereas $ΔNTE^O$ was allowed to move freely, no other restraints were applied. The fitting procedure showed high convergence ($χ^2$ for all 20 models generated were close to 1); however, most of the models could be discarded because they contradicted biochemical data. The resulting model of the complex was free from clashes and consistent with all accumulated experimental information, including the disulfide-linked pairs used in this work. The resulting topology was supported by the distribution of the electrostatic potentials on the surface of proteins calculated individually for FRP and $ΔNTE^O$ using APBS plugin for PyMOL[64], and by the conservativity analysis for the FRP dimer performed using Consurf[65] (fifty FRP homologs from different cyanobacteria were taken[25]). Superposition of the atomistic model with the best-fitting GASBOR-derived[66] ab initio model ($χ^2 = 1.01$; CorMap 0.351) calculated directly from the SAXS data resulted in an NSD value of 1.85. Models of individual $ΔNTE^O$ or the oxFRPcc dimer with supplemented flexible residues could not describe the SAXS data for the 1:2 complex and provided inadequate fits ($χ^2 = 22$ and 41, respectively). Structural models were drawn in PyMOL.

**Absorption spectroscopy.** Steady-state absorption spectra and time-courses of absorption were recorded using a setup including Maya2000 Pro spectrometer (Ocean Optics, USA) and a stabilized broadband fiber-coupled light source (SLS201L/M, Thorlabs, USA). Temperature of the samples in 10 mm quartz cuvettes was stabilized by a Peltier-controlled cuvette holder Qpod 2e (Quantum Northwest, USA) with a magnetic stirrer. A 900 mW blue light-emitting diode (M455L3, Thorlabs, USA), with a maximum emission at 455 nm was used for $OCP^O → OCP^R$ photoconversion of the samples. Light-induced accumulation of $OCP^R$ is reversible due to the spontaneous or FRP-mediated $OCP^R → OCP^O$ back-conversion, which is considered to be light-independent. The kinetics of OCP photoinduced transitions was measured with 100 ms time resolution as the change of optical density at 550 nm, since the most noticeable changes in OCP absorption occur in this spectral region. Under constant illumination by actinic light, OCP samples and OCP/FRP mixtures exist in equilibrium between the red and orange states, which in the absence of FRP is shifted towards the red state. Amplitudes of photoconversion were estimated under actinic light as maximal changes in optical density at 550 nm comparing to the dark-adapted state. Time-courses of $OCP^R → OCP^O$ back-conversion were approximated by decaying exponential function in order to estimate characteristic lifetimes and rates. The experiments were repeated three times using different protein preparations and the most typical results are presented.

## Data availability

Structural models and SAXS profiles are deposited with the SASBDB[67] under accession codes SASDDE9, SASDDF9, and SASDDG9. All other data supporting the findings of this study are available from the corresponding author upon reasonable request.

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

## Acknowledgements

N.N.S. is grateful to Prof. Sergei Strelkov, Dr. Oleg Klychnikov (Biocrystallography Lab, KU Leuven) and to Prof. Alfred Antson (YSBL, UK) for fruitful discussions, to Dr. Cy Jeffries (EMBL Hamburg) and Oleg Zakharchenko (Lomonosov MSU) for help with SAXS measurements, and to iNEXT for supporting the SAXS-683 experiment (iNEXT proposal PID: 2977). The authors acknowledge the support of the Russian Science Foundation (Grant no. 18-44-04002), the German Ministry for Education and Research (BMBF grant no. 01DJ15007) and the German Research Foundation (DFG grant no. FR1276/5-1). The study was partially supported by the grant from the Russian Foundation for Basic Research (No. 18-04-00691 to N.N.S.). Protein–protein interactions were studied in the framework of the Program of the Federal Agency of Scientific Organizations (No. 0104-2018-0001). E.A.S. was supported by the Russian Foundation for Basic Research (No. 16-05-01110).

## Author contributions

N.N.S. conceived the idea and supervised the study; N.N.S., Y.B.S., and E.G.M. designed the experiments; N.N.S., Y.B.S., M.M., and E.G.M. performed the experiments; N.N.S. performed SAXS data collection and analysis; N.N.S., E.A.S., T.F., and E.G.M. analyzed the data; N.N.S. drew the figures; N.N.S. wrote the paper with an input from Y.B.S., E.A.S., T.F., and E.G.M.

## Additional information

**Competing interests:** The authors declare no competing interests.

