## [Peer Review File · Nature Communications]

Reviewers' comments:

Reviewer #1 (Remarks to the Author):

In this work, the authors study the interaction between the cyanobacterial Orange Carotenoid Protein (OCP) and the Fluorescence Recovery Protein (FRP). Both proteins are involved in a photoprotective NPQ mechanism which by increasing heat dissipation of excess absorbed energy at the level of the phycobilisome (cyanobacterial antenna), decreases the energy arriving at the reaction centers. The OCP is a modular photoactive protein composed by two domains (CTD and NTD) that share a keto-carotenoid molecule. In darkness, it is orange and has a closed conformation. Upon illumination with strong light, conformational changes of the carotenoid and the protein are induced, the domains are separated and the carotenoid translocated into the NTD. The photoactivated protein becomes red and it is the active form. The carotenoid-NTD interacts with the phycobilisome inducing energy and fluorescence quenching. The FRP is needed to detach the OCP from the phycobilisome and to accelerate the reconversion of the red form into the orange inactive form.

The authors have already done important contributions in order to elucidate the interaction between the FRP and the OCP and the activity of the FRP. They were the first to present evidences strongly suggesting that the FRP, that is a dimer (dFRP) in solution, monomerizes (mFRP) upon binding to the CTD of the OCP red. However, a study using cross-linking from Blankenship's group showed the presence of high concentrations of CTD-dFRP complexes suggesting that monomerization occurs later if it occurs. The questions that remained to be answered were: when this monomerization occurs if it occurs (this must be confirmed) and if the monomerization is essential for the FRP activity. The authors try to answer to these two questions in the present work. Based in their results they present a model for the action of the FRP.

The authors had the excellent idea of constructing two FRP mutants. One of them it is always a monomer and the other one it is always a dimer in oxidizing conditions. Nevertheless, the second mutant can monomerizes under reducing conditions.

First, the authors realized several experiments in order to confirm that one FRP mutant (L49E FRP) is a monomer that could be compare to the WT monomer and the second one (FRPcc), containing a cys in each monomer, is a dimer that under reducing conditions can monomerize. The results are convincing.

Then, they did some experiments to further characterize the FRP mutants.

1) Circular dichroism: It supports the idea that the secondary structures of WT FRP and FRPcc mutant are similar. In addition, it indicates that the alpha-helix content of the L49E FRP monomer is reduced (from 60% to 40%). This suggests that monomerization of FRP is accompanied by local unfolding of the protein. This was also previously suggested by results obtained in Blankenship's laboratory (Lu et al, 2017)

2) Intrinsic Trp fluorescence: These experiments confirmed that the L49E FRP mutant is a monomer in which the phase of interaction with the other monomer in the dimer is solvent exposed. In addition, the experiments confirmed that the FRPcc dimer remains as a dimer under oxidizing conditions but it can monomerize as the WT FRP under reducing conditions.

3) SAXs: I am not expert in this technique and I cannot be critical about the results. Nevertheless, they seem to confirm that FRPcc and FRP WT have similar secondary conformation as dimers and that FRP L49E is a monomer with a conformation different to that of the monomer in the dimer. This different conformation seems not to be associated to the mutation but to the properties of a FRP monomer. This is a very important result and in order to elucidate the activity of FRP will be essential to resolve its structure in a future.

Once demonstrated that FRPcc is a dimer that cannot monomerize under oxidizing conditions and that L49E FRP is a monomer, the authors studied the interaction of these FRP mutants with the OCP.

FRP binds with a high affinity only to OCPred. It first binds to the CTD and then it seems that it interacts also with the NTD and brings both domains together. FRP has a very low affinity for OCPo. In the past the authors showed that: 1) FRP can permanently bind to an OCP mutant that is continuously in the red form, OCPaa; 2) FRP can also bind to an orange OCP that lacks the N-terminal arm (NTE) and finally, 3) FRP also binds to the dimeric CTD.

In this work, the authors show that the dimeric FRPcc mutant was also able to bind to all these OCPs not only under reducing conditions but also under oxidizing conditions in which it was not able to monomerize. Figure S4 that demonstrates this fact must be shown as a figure in the text. Moreover, this dimeric FRP is active even without monomerization.

By contrast, the monomer L49E FRP mutant was unable to bind to any of these proteins. These results are very convincing and clearly demonstrated that FRP binds to the OCPred as a dimer and

that if it monomerizes this monomerization occurs after the binding. These results also strongly suggested that the dimer interface of the FRP is not involved in the binding as already suggested by results of Blankenship's (Lu et al, 2017) and Kirilovsky's laboratory (Thurotte et al, 2016). The authors have now a problem. In their previous article, they showed several experiments (SEC and non-denaturant gels) that for them, they demonstrated that a monomer of FRP and not a dimer was attached to the OCPaa. Here, in supplementary data, they show other experiments using FRPs from *Arthrospira*, *Anabaena* and *Synechocystis* to show this again. I am very sorry, but I am not convinced by the experiments (at least not in the way the results are shown). To convince me and the other readers, the authors have to show in the same figure (and with the same scale) the SEC position of a complex containing OCP-dFRP and a complex containing OCP-mFRP. The position of the peaks must be different since the MW of OCP^{red}-mFRP is 45 kDa and that of OCP-dFRP is 60 kDa. They show a figure similar to this in Fig 4D but they use *Arthrospira* FRP instead of *Syn* FRP. Why? Please show the figure with *Syn* WT FRP. This is the best demonstration for the existence of OCP-mFRP complexes.

The cross-linking experiments are not convincing to demonstrate that in one case is a monomer and that in the other case is a dimer. In the presence of glutaraldehyde the FRP dimer is only slightly stabilized (see line 2 in Figure 4A). Thus, it is possible that although a dimer is attached to the OCP, glutaraldehyde stabilizes the interaction OCP-mFRP and very little the interaction FRP-FRP that is destroyed by SDS. I agree with the authors that their results suggest at least a partial monomerization but they are not completely demonstrating this. The SEC experiment will show this more clearly.

One important question: Why you did the experiments with the deltaNTE orange that has a "closed" form instead that with the delatNTE red or OCPaa that both have an OCP red like structure. FRP binds to OCPred and not to a "closed" OCP? The results could be different if you use OCPaa. Please explain your choice. Seeing the working model in Figure 8, I could understand the choice. However, this must be explained earlier in the article and the experiments with the red form must be also done.

The authors also observed the formation of complexes of around 91 kDa that they deduced that are complexes containing 2 OCP and one FRP dimer. These experiments were done with deltaNTE (orange) and with glutaraldehyde. Are these complexes formed with OCPaa and in the absence of glutaraldehyde? Based on these results, the authors proposed as a working hypothesis that monomerization of FRP occurs once two OCPs attach an FRP dimer. When this occurs? When the OCP are still red and open or then when they are closed?

I am sorry but I am not sure if I well understand what the authors show in Figure 5 A. Are you showing the binding of dFRP to one OCP? And then do you compare the binding of dFRPcc with that of WT FRP showing that they are similar? In this case, what do you conclude? It is not clear for me. I conclude that in both cases a dFRP binds to the OCP and only after a time there is monomerization (if there is) following one or the other of the ways shown in supplementary text 1. But from this experiment we cannot decide what way is the good one. Please explain more clearly what you think and interpret these results.

Then, the authors did experiments to demonstrate that this is what occurs in vivo: Monomerization of FRP upon binding of two OCP molecules to one FRP dimer.

This idea is tempting but I think that this hypothesis although possible, it is not confirmed based on the results shown in this article. I will explain why I think this.

1) FRP interacts with OCPred (open form) and the authors did SAXS and all the structural analysis using deltaNTE OCP (orange). Although, FRP can attach to this OCP mutant (without necessity of photoactivation) since the NTE is absent, for me it is not a good model for FRP binding to the CTD in OCPred. Also it is not a good model to study the binding of the FRP to OCPorange because it lacks the NTE that in the WT hinders FRP binding. Thus, all the results obtained and the models derived from SAXS are right (and I believe them) for the FRP binding to deltaNTE OCP orange. I suppose that the authors used this OCP mutant because the "mobile" OCPred conformation could difficult SAXS models. Again, please explain your choice.

2) The FRP and OCP amino acids involved in the interaction are in agreement with previous results from Blankenship's laboratory (FRP K102 and F76) and from Kirilovsky's lab (OCP F299) that used OCP red forms. Thus, I agree that these amino acids could be involved in both FRP binding to OCPred and deltaNTE OCP orange. Moreover, they confirm the interaction between OCP F299 and FRP K102 using F2999C and F76C mutants that form an S-S bond under oxidizing conditions and the complex appeared as a 45kDa band in an SDS gel. Very nice experiment. These are important results.

3) Then, the authors show that the binding of two deltaNTE OCP to an FRP dimer causes an apparent clash between the NTD of both OCPs. However, this clash will not be produced with OCPred forms in which the positions of NTDs will be completely different. Thus, I do not agree with the hypothesis that this clash could be the cause of monomerization of FRP. This can occur only

when the OCP are again orange. In this case, monomerization is not helping FRP activity since occurs after. (Please see also my comments about Fig 8. In the figure the ideas of authors are more clearly explained. This explanation must arrive before)

In conclusion of this part: I agree that in OCPred, the CTD and FRP interaction involves FRP K102 and F76 and OCP299. By contrast, I do not agree with the idea of the clash of red OCPs when two OCPs bind to a FRP dimer. Secondly, the models shown in Figure 6A and B are right for deltaNTE OCP orange but not for WT OCPred (open domains) or OCP orange (presence of NTE, avoiding binding of FRP). Thus, if there is a clash is once the OCP becomes orange again. Why this could be useful?

Then, the authors compared the FRP activities of the WT FRP dimer (that is able to monomerize) and FRPcc (oxidizing conditions) that is unable to monomerize. The experiments show that WT FRP is more effective to accelerate OCPred to OCPorange than FRPcc suggesting that the monomerization of FRP (although not essential) increases FRP effectiveness.

Figure 8: The model: The model is OK because all the possibilities are open. In the model also are more clear the ideas of the authors about how and when the second OCPred could bind the FRP dimer and shows that the clash could occur once both OCPred were "closed" by FRP action. This is based on the model of clash shown in Figure 5E. I am still think that the binding of two OCP to an FRP dimer must be rare to occur in the cells but there is a possibility that occurs in vitro under the experimental conditions used. My idea is that after FRP dimer binding to the CTD, the interaction of the FRP monomers is weakened and the interface domain including R60,D54 and others amino acids important for FRP activity will be completely free to interact with the NTD and completely close the OCP. On the other hand, monomerization could occur after the domains will be closed (or partially closed). What it is sure that this monomerization helps the FRP activity. Nevertheless, the authors can have different ideas and prefer the "clash" hypothesis. All the time that they do not say that they demonstrated this hypothesis is OK for me.

In conclusion, this work present interesting results that allow a better understanding of FRP-OCP interaction and FRP activity. It is an important work that only need some more clear figures and more early and clear explanations to reinforce the message. Please, also add the experiments with the red OCPs interacting with FRP. If there is a reason why these experiments cannot be done or they will not give more information, please explain.

I hope that my comments will help you to improve your article that presents important findings and ideas that will largely influence the thinking of the field. I think that changing a little the presentation of figures and explaining better your ideas you will convince everybody that FRP monomerizes and which are the amino acids that are involved in the FRP-OCP interaction. About the mechanism of monomerization, I am less convinced but your results allow you to propose that monomerization can occurs once the OCP is again "close". All the time that this remains as a working model, it is OK for me.

Diana Kirilovsky

Reviewer #2 (Remarks to the Author):

In their manuscript "Topology of the OCP/FRP complexes explains the molecular mechanism controlling high light tolerance in cyanobacteria" Sluchanko et al. characterize the interactions of the fluorescence recovery protein FCP with orange carotenoid protein OCP using a variety of biophysical methods, including SAXS.

They could derive a SAXS based model of the Δ NTE(O)/ocFRPcc complex which explains the absence of 2:2 complexes due to steric hindrance and implies a mechanism for the creation of 1:1 complexes in the case FRPwt. This models is also in good agreement with crosslinking mass-spectrometry results.

Surprisingly, the authors seem to have calculated the theoretical SAXS curves of the models with CRY SOL instead directly using the fits created by CORAL. If there was a discrepancy between these curves, it should be addressed.

Despite listing the most important parameters (including the shape of the Kratky plot), the presentation of the SAXS results does not follow the guidelines for SAS based modeling of macromolecules by the IUCr (<http://journals.iucr.org/d/issues/2017/09/00/jc5010/>). It would be of particular interest to see the residuals of the fits in figures 5 and S3 as well as the pair distribution functions used for GASBOR modeling, as well as the number of beads used for fitting. For the Mass based on the Porod volume a citation for the proportionality factor needs to be provided. Citations are also missing for the $V_{_c}$ and MoW methods to derive the mass.

On page 20, line 650 the following citation is missing:

Blanchet, C.E., Spilotros, A., Schwemmer, F., Graewert, M.A., Kikhney, A.G., Jeffries, C.M., Franke, D., Mark, D., Zengerle, R., Cipriani, F., Fiedler, S., Roessle, M. and Svergun, D.I (2015) Versatile sample environments and automation for biological solution X-ray scattering experiments at the P12 beamline (PETRA III, DESY) J. Appl. Cryst. 48(2)

Finally, I have a few more "linguistic" comments: I sincerely hope that the authors did not measure any hydrodynamic parameters in their SAXS experiment as that would imply that wither the protein complex affected the flow of the sample or the flow of the sample affected the complex. "Structural parameters in solution" would be more accurate. And "Rg (reverse)" in the SAXS-tables should read " R_{g} (reciprocal)".

Reviewer #3 (Remarks to the Author):

This paper describes the combination of genetical engineering of FRP samples of specific oligomerization state with various biophysical and biochemical techniques in order to unravel the mechanism of FRP binding to OCP. The latter is an important process in cyanobacterial photoprotection. The study shows that OCP preferentially binds dimeric FRP, which only subsequently monomerizes. The results allow for identification of potential FRP binding sites. The results are new and original and significantly advance knowledge in a dynamic field of high scientific interest. Therefore, it is of interest to a broad readership. The paper can be recommended for publication, if the following minor issues can be resolved:

- 1.) page 3: ".. permitting rational engineering of the oligomeric state." If the presence of the dimeric form was already shown by SAXS, what type of engineering is necessary at this point? If this is not meant, the phrase "oligomeric state" should be more properly defined.
- 2.) page 3, bottom: the statement "Under these conditions..." is unclear. Before, the authors talk about a monomeric form, in what follows about the dimeric form. Which similar conditions are described or inferred here?
- 3.) page 4, top: "at highest protein concentration": A dependence on concentration has not been discussed up to this point. It has to be specified, what "highest protein concentration" is.
- 4.) page 6, bottom: "The SAXS profiles... were averaged and the resulting rather noisy curve...": It can only be deduced by coincidental reading of the supplementary material that 4 data sets of supposedly identical samples were averaged plus one data set of a different concentration. First, the main text should be consistent in itself and readable without the supplementary material. Second, the procedure is not understandable and seems to suggest a heterogeneity among the samples. How can this be explained?
- 5.) page 7, middle: The form OCP^{AA} has not been introduced beyond the entry in the list of abbreviations. The reader is forced to guess that it is an equivalent of the active form OCPR.
- 6.) page 7, middle: "Carotenoid-bound ... CTD" sounds wrong
- 7.) page 10/11: the major question along with this manuscript is, why the complex topology was studied only using the deltaNTE-form of OCP, if OCP^{AA} was available and seems to resemble the active state OCPR ?
- 8.) The study mainly relies on SAXS data. Can the authors exclude radiation damage and why?

Minor comments:

page 3, middle: correct ".. two of its homologs..."

page 7, middle: "whereas already bound FRP monomer...", article seems to be missing

Reviewer #1 (Remarks to the Author):

In this work, the authors study the interaction between the cyanobacterial Orange Carotenoid Protein (OCP) and the Fluorescence Recovery Protein (FRP). Both proteins are involved in a photoprotective NPQ mechanism which by increasing heat dissipation of excess absorbed energy at the level of the phycobilisome (cyanobacterial antenna), decreases the energy arriving at the reaction centers. The OCP is a modular photoactive protein composed by two domains (CTD and NTD) that share a keto-carotenoid molecule. In darkness, it is orange and has a closed conformation. Upon illumination with strong light, conformational changes of the carotenoid and the protein are induced, the domains are separated and the carotenoid translocated into the NTD. The photoactivated protein becomes red and it is the active form. The carotenoid-NTD interacts with the phycobilisome inducing energy and fluorescence quenching. The FRP is needed to detach the OCP from the phycobilisome and to accelerate the reconversion of the red form into the orange inactive form.

The authors have already done important contributions in order to elucidate the interaction between the FRP and the OCP and the activity of the FRP. They were the first to present evidences strongly suggesting that the FRP, that is a dimer (dFRP) in solution, monomerizes (mFRP) upon binding to the CTD of the OCP red. However, a study using cross-linking from Blankenship's group showed the presence of high concentrations of CTD-dFRP complexes suggesting that monomerization occurs later if it occurs. The questions that remained to be answered were: when this monomerization occurs if it occurs (this must be confirmed) and if the monomerization is essential for the FRP activity. The authors try to answer to these two questions in the present work. Based in their results they present a model for the action of the FRP.

The authors had the excellent idea of constructing two FRP mutants. One of them it is always a monomer and the other one it is always a dimer in oxidizing conditions. Nevertheless, the second mutant can monomerizes under reducing conditions.

First, the authors realized several experiments in order to confirm that one FRP mutant (L49E FRP) is a monomer that could be compare to the WT monomer and the second one (FRPcc), containing a cys in each monomer, is a dimer that under reducing conditions can monomerize. The results are convincing.

Then, they did some experiments to further characterize the FRP mutants.

1) Circular dichroism: It supports the idea that the secondary structures of WT FRP and FRPcc mutant are similar. In addition, it indicates that the alpha-helix content of the L49E FRP monomer is reduced (from 60% to 40%). This suggests that monomerization of FRP is accompanied by local unfolding of the

protein. This was also previously suggested by results obtained in Blankenship's laboratory (Lu et al, 2017)

2) *Intrinsic Trp fluorescence*: These experiments confirmed that the L49E FRP mutant is a monomer in which the phase of interaction with the other monomer in the dimer is solvent exposed. In addition, the experiments confirmed that the FRPcc dimer remains as a dimer under oxidizing conditions but it can monomerize as the WT FRP under reducing conditions.

3) *SAXs*: I am not expert in this technique and I cannot be critical about the results. Nevertheless, they seem to confirm that FRPcc and FRP WT have similar secondary conformation as dimers and that FRP L49E is a monomer with a conformation different to that of the monomer in the dimer. This different conformation seems not to be associated to the mutation but to the properties of a FRP monomer. This is a very important result and in order to elucidate the activity of FRP will be essential to resolve its structure in a future.

Once demonstrated that FRPcc is a dimer that cannot monomerize under oxidizing conditions and that L49E FRP is a monomer, the authors studied the interaction of these FRP mutants with the OCP.

FRP binds with a high affinity only to OCPred. It first binds to the CTD and then it seems that it interacts also with the NTD and brings both domains together. FRP has a very low affinity for OCPo. In the past the authors showed that: 1) FRP can permanently bind to an OCP mutant that is continuously in the red form, OCPaa; 2) FRP can also bind to an orange OCP that lacks the N-terminal arm (NTE) and finally, 3) FRP also binds to the dimeric CTD.

Thank you very much for a very deep understanding of our results and a high mark to our contribution, which is especially nice to receive from the outstanding specialist in the field.

In this work, the authors show that the dimeric FRPcc mutant was also able to bind to all these OCPs not only under reducing conditions but also under oxidizing conditions in which it was not able to monomerize. Figure S4 that demonstrates this fact must be shown as a figure in the text. Moreover, this dimeric FRP is active even without monomerization.

Although we agree that these data are important, we left the figure in the supplement for two reasons. First, these experiments were done in slightly different conditions and therefore would not be a good addition to the current Fig. 3, while we have no more slots to present these results as a separate figure (currently 8 main figures and 2 tables in the main text; ten allowed items in total). The second reason was that we wanted to avoid repetitions as the interaction of OCP with the oxidized FRP dimer is also discussed in more detail and from different perspectives throughout the rest part of the paper, including structural analysis in solution. Please excuse our choice.

By contrast, the monomer L49E FRP mutant was unable to bind to any of these proteins. These results are very convincing and clearly demonstrated that FRP binds to the OCPred as a dimer and that if it monomerizes this monomerization occurs after the binding. These results also strongly suggested that the dimer interface of the FRP is not involved in the binding as already suggested by results of Blankenship's (Lu et al, 2017) and Kirilovsky's laboratory (Thurotte et al, 2016).

The authors have now a problem. In their previous article, they showed several experiments (SEC and non-denaturant gels) that for them, they demonstrated that a monomer of FRP and not a dimer was attached to the OCPaa. Here, in supplementary data, they show other experiments using FRPs from *Arthrospira*, *Anabaena* and *Synechocystis* to show this again. I am very sorry, but I am not convinced by the experiments (at least not in the way the results are shown). To convince me and the other readers, the authors have to show in the same figure (and with the same scale) the SEC position of a complex containing OCP-dFRP and a complex containing OCP-mFRP. The position of the peaks must be different since the MW of OCP⁺-mFRP is 45 kDa and that of OCP-dFRP is 60 kDa. They show a figure similar to this in Fig 4D but they use *Arthrospira* FRP instead of *Syn* FRP. Why? Please show the figure with *Syn* WT FRP. This is the best demonstration for the existence of OCP-mFRP complexes.

We are sorry for not clearly explaining that in the original paper, this is now corrected in the revised version. The situation is that we do observe monomerization of all FRPs, but this process is not fully controlled and seems to be dependent on a concentration regime (e.g., molar ratios OCP/FRP and absolute concentrations) in the case of SynFRP and AnaFRP, but not in the case of AmaxFRP, which for some reason demonstrates almost exclusively 1:1 complexes with various OCP forms that it can bind. We believe that 1:2 and 1:1 complexes represent intermediate steps of the interaction and, in the case of AmaxFRP, the 1:1 intermediate is relatively more populated than in others (which may also contain the 1:2 intermediate). For these reasons, we used AmaxFRP as a reference of the 1:1 complex, for example, for its position on the SEC profile. For SynFRP and AnaFRP, we most of the time see a mixture of the 1:1 and 1:2 complexes, and this seems to be related to the different stability of their dimeric interfaces. Such a comparative analysis was published in Slonimskiy et al, BBA, 2018. In other words, AmaxFRP is taken as an “edge” state, which in particular clarified that the position of the 1:1 and 1:2 complexes (with oxFRPcc) is very similar on the SEC profile (indeed, as the reviewer pointed out, it is in Fig. 4D). This is most likely because of the differences in the shape of the two complexes since SEC separates particles not by mass, but by hydrodynamic size, which is related to the shape. In the previous experiments (Sluchanko et al BBA 2017, Sluchanko et al FEBS Lett 2017), the monomerization was detected by a combination of methods, but we didn't claim that the FRP monomerization was always complete. In the present study, by using dissociation-incapable FRP we could distinguish between 1:1 (AmaxFRP) and 1:2 complexes (oxFRPcc) and understand the mechanism in much more detail. We showed that, alongside spontaneous monomerization of the OCP-bound FRP, some active mechanism may take place, i.e., via a 2:2 binding that we showed here by chemical crosslinking. Anyway, we have added more explanations to the text and also clarified the presentation of our results, which we hope are really convincing.

The cross-linking experiments are not convincing to demonstrate that in one case is a monomer and that in the other case is a dimer. In the presence of glutaraldehyde the FRP dimer is only slightly stabilized (see line 2 in Figure 4A). Thus, it is possible that although a dimer is attached to the OCP, glutaraldehyde stabilizes the interaction OCP-mFRP and very little the interaction FRP-FRP that is destroyed by SDS. I agree with the authors that their results suggest at least a partial monomerization but they are not completely demonstrating this. The SEC experiment will show this more clearly.

First of all, chemical crosslinking gives only partial crosslinking all the time, and thus it is OK to have only partial fixation of the FRPwt dimer (Fig. 4A) even despite in solution there are only dimers. This is related to the incomplete crosslinking efficiency and also to the limited amount of crosslinkable residues in the interface of SynFRP, this is why we also compared GA crosslinking with the Anabaena homolog of FRP, which has four Lys residues in the interface (discussed in the paper), but this showed very limited amount of the 1:2 OCP:FRP complexes as well, suggesting at least partial monomerization of FRP sitting on OCP. As said above, on SEC, 1:1 and 1:2 complexes have very close positions due to different hydrodynamic properties and we seem to have mostly 1:1 but also 1:2 complexes due to the spontaneous but incomplete FRP monomerization. As mentioned, it also depends on protein concentrations and ratios used. In any case, the phenomenon of FRP monomerization was studied in detail in our previous works and, in particular, it was clearly shown by quantification of the OCP and FRP bands forming the complex on SEC (Sluchanko et al BBA 2017) and by mixing proteins with different FRP excess (Slonimskiy BBA 2018), but seems to be deepest in the AmaxFRP case, which is taken in our present work as a most “clear” reference for the 1:1 complexes. We have tried to correct the text accordingly to clarify these points.

One important question: Why you did the experiments with the deltaNTE orange that has a “closed” form instead that with the delatNTE red or OCPaa that both have an OCP red like structure. FRP binds to OCPred and not to a “closed” OCP? The results could be different if you use OCPaa. Please explain your choice. Seeing the working model in Figure 8, I could understand the choice. However, this must be explained earlier in the article and the experiments with the red form must be also done.

That is indeed a very important question (which was also raised by the Reviewer 3). We have no doubts that, under physiological conditions, the productive interaction of dFRP and OCP begins when OCP domains are separated, and this state represents an active quenching form. The problem here is that the conformational mobility of the active form is very high, since domains are connected only by a flexible

linker, so one can imagine multiple structures of OCP(red)-FRP complexes which appear immediately after the dFRP binding: in other words, this system is characterized by the high conformational heterogeneity. Since binding is due to interactions of FRP with the primary site in CTD, we postulate that initial position of NTD is not that important, but its mobility, of course, will bring substantial uncertainty to any structural method, especially SAXS. However, since the main activity of FRP is to assist proper (OCP orange-like) alignment of the OCP domains, one of our goals was to study how dFRP could be positioned in a complex when its job is (almost) done. From the other hand, it is generally accepted that FRP does not bind to OCP in its basal orange state (or does it in a different mode and with low affinity), so after the domain positioning is finished, FRP must dissociate from OCP. Since our idea was to trap an intermediate of the OCP-FRP interactions, we had to deal with these issues (conformational flexibility of the active red state, low binding efficiency in the basal orange OCP state), so our choice of the deltaNTE mutant as a model object was dictated by two main reasons - (1) domains are already (almost perfectly) aligned (even without the FRP assistance), but (2) binding of FRP in CTD is very strong since the absence of NTE exposes the primary site (Sluchanko et al FEBS Lett 2017). It is very important to note that due to the (proven) structural similarity of the deltaNTE mutant and OCP in its orange state (Fig. S3B), we were able to use the X-ray structure of OCP to constrain the deltaNTE/dFRPcc interactions in solution and obtain the structural model based on SAXS. Regarding the experiments with the active OCP species, considering the structural heterogeneity of the active red OCP, the SAXS data for OCPR (and any other species with separated domains) represent an ensemble of conformations that is difficult to take into account during modeling (or is done with decreased certainty). The situation gets even more complicated due to the tendency of activated OCPs to form dimers at concentrations which are necessary for SAXS measurements ensuring low noise (Maksimov et al Biophysical Journal 2017). Thus, the evaluation of the SAXS data for OCPR-dFRPcc or OCPaa-dFRP would require consideration of the substantial structural heterogeneity, which immediately would reduce the overall resolution and reliability of the models. Moreover, we are sure that FRP does not care about the presence of carotenoid in OCP, but this factor determines stabilization of the OCP structure after spatial alignment of the domains. Unfortunately, mutation(s) of Trp-288 or Tyr-201, mimicking the photoactivated state, not only prevent deactivation of OCP but also reduce(s) efficiency of carotenoid binding, so all preparations of mutants represent a mixture of their apo and holo forms, which although both can interact with FRP, definitely have different lifetimes of the states with the proper domains positioning, further increasing heterogeneity. Although we did SAXS experiments with models of constantly active OCPs and FRPs, considering abovementioned issues, we are not ready to give a straightforward interpretation of such complex structures in solution. This may be proven in the future, but we assume that the most probable structure of active OCP-dFRP complex which appears immediately after binding could be obtained on the basis of our SAXS data on deltaNTE-dFRP, in which (mutual) positions of CTD and dFRP are preserved, while the NTD is able to move freely, reaching position proposed in Gupta et al. PNAS 2015. At the same time, our data describe the important intermediate of the binding process (1:2) and provide a reliable basis for modeling.

The authors also observed the formation of complexes of around 91 kDa that they deduced that are complexes containing 2 OCP and one FRP dimer. These experiments were done with deltaNTE (orange) and with glutaraldehyde. Are these complexes formed with OCPaa and in the absence of glutaraldehyde? Based on these results, the authors proposed as a working hypothesis that monomerization of FRP occurs once two OCPs attach an FRP dimer. When this occurs? When the OCP are still red and open or then when they are closed?

While our experiments with glutaraldehyde demonstrate that 2:2 complexes could be covalently trapped, we agree that the probability (and rather, stability) of such complexes formation under physiological conditions is rather low, in line with a short lifetime in complex with FRP and steric hindrances. Indeed, to get 2:2 complexes, binding of the second OCP to dFRP must occur before either the OCP domains are aligned (as after that the steric clash takes place) or the dFRP dissociates. Moreover, in the wtOCP-dFRP complex the NTE may facilitate detachment of FRP from OCP, thus also decreasing the effective lifetime of the dFRP-OCP complex and, hence, the probability of 2:2 complex formation.

Next, we have no evidence that FRP binding/detachment influences the color of the chromophore since FRP monomerization happens even upon the interaction with the OCP apoform, so we assume that carotenoid only increases the lifetime of the “closed” state in which FRP could be detached.

Overall, we don't think that 2:2 binding is a major reason of FRP monomerization, but since we observed a principle possibility for such complex formation, we cannot exclude that it will provoke FRP monomerization in a real situation, for example, when the (local) OCP concentrations are very high. Nevertheless, the crosslinking experiments with OCPaa were done and showed the principal ability of the formation of the 2:2 complexes as in the deltaNTE case (Fig. S7). Under equilibrium conditions, i.e. in the absence of crosslinking, the 2:2 complexes **could be** detected on SEC using very high load concentrations of OCP and FRP, but even in this case, the relative amount of these complexes was very low. Again, our understanding is that FRP can spontaneously dissociate in the course of interaction with OCP, but the increased OCP concentrations can further provoke the monomerization by splitting bigger complexes into 1:1 subcomplexes.

I am sorry but I am not sure if I well understand what the authors show in Figure 5 A. Are you showing the binding of dFRP to one OCP? And then do you compare the binding of dFRPcc with that of WT FRP showing that they are similar? In this case, what do you conclude? It is not clear for me. I conclude that in both cases a dFRP binds to the OCP and only after a time there is monomerization (if there is) following one or the other of the ways shown in supplementary text 1. But from this experiment we cannot decide what way is the good one. Please explain more clearly what you think and interpret these results.

SEC data suggest that dFRP and dFRPcc can bind to delatNTE OCP, taken at low protein concentration, with almost identical apparent Kd; this means that the introduced mutations do not affect the (primary) binding interface, the primary binding step, and, thus, the binding efficiency. This result confirms that the initial binding is the recruitment of the FRP dimer to OCP monomer. Indeed, monomerization occurs after the binding and this is the main reason why WT FRP and FRPcc have different rates of OCP_R to OCP_O conversion. Considering the fact that FRPcc still accelerates back conversion, we can postulate that monomerization is not necessary, however, it gives some advantages since it allows for significantly faster FRP cycling, which means that lower concentration of FRP would be necessary to prevent PBs quenching in vivo. The reviewer is right that we can't conclude that 2:2 binding has any advantages in monomerization comparing to the process which happens upon 1:2 binding, only as a factor stimulating (spontaneous) FRP monomerization.

Then, the authors did experiments to demonstrate that this is what occurs in vivo: Monomerization of FRP upon binding of two OCP molecules to one FRP dimer.

We did not present experiments confirming that 2:2 binding happens in vivo. However, our structural models and the results of chemical crosslinking support this principal ability that provides an interesting mechanistic insight into the FRP function and the topology of its complexes with OCP, which was completely enigmatic previously.

This idea is tempting but I think that this hypothesis although possible, it is not confirmed based on the results shown in this article. I will explain why I think this.

1) FRP interacts with OCPred (open form) and the authors did SAXS and all the structural analysis using deltaNTE OCP (orange). Although, FRP can attach to this OCP mutant (without necessity of photoactivation) since the NTE is absent, for me it is not a good model for FRP binding to the CTD in OCPred. Also it is not a good model to study the binding of the FRP to OCPorange because it lacks the NTE that in the WT hinders FRP binding. Thus, all the results obtained and the models derived from SAXS are right (and I believe them) for the FRP binding to deltaNTE OCP orange. I suppose that the authors used this OCP mutant because the “mobile” OCPred conformation could difficult SAXS models. Again, please explain your choice.

Due to the transient and metastable nature of OCP_R, the FRP-OCP complex is not kinetically stable under physiological conditions by definition, so it is not completely correct to call any of models representing the steady state good or bad. Since our idea was to trap an intermediate of the OCP-FRP

interactions (not orange OCP-FRP or red OCP-FRP), we had to deal with the conformational flexibility of the active red state (the reviewer's supposition is absolutely right!) and low binding efficiency in the basal orange OCP state, so our choice of the deltaNTE mutant as a model object was dictated by two main reasons discussed above. We don't agree that the absence of NTE may somehow decrease the relevance of the binding site in the CTD since, in the red state, the NTD is far away from the CTD, that was nicely demonstrated by others (Gupta et al. PNAS 2015) and in our works. We agree that the NTE prevents FRP binding in the orange state, but when domains are separated it is likely not involved in the interaction with FRP, so its function becomes relevant only after the domains are aligned (and most probably FRP is spontaneously detached). So our model (based on the deltaNTE mutant and constantly dimeric FRP) represents an important intermediate of the photocycle in which domains are already aligned, but FRP is still attached, so the NTE could be positioned anywhere since it is connected to NTD by a flexible loop.

2) *The FRP and OCP amino acids involved in the interaction are in agreement with previous results from Blankenship's laboratory (FRP K102 and F76) and from Kirilovsky's lab (OCP F299) that used OCP red forms. Thus, I agree that these amino acids could be involved in both FRP binding to OCPred and deltaNTE OCP orange. Moreover, they confirm the interaction between OCP F299 and FRP K102 using F2999C and F76C mutants that form an S-S bond under oxidizing conditions and the complex appeared as a 45kDa band in an SDS gel. Very nice experiment. These are important results.*

Thank you!

3) *Then, the authors show that the binding of two deltaNTE OCP to an FRP dimer causes an apparent clash between the NTD of both OCPs. However, this clash will not be produced with OCPred forms in which the positions of NTDs will be completely different. Thus, I do not agree with the hypothesis that this clash could be the cause of monomerization of FRP. This can occur only when the OCP are again orange. In this case, monomerization is not helping FRP activity since occurs after. (Please see also my comments about Fig 8. In the figure the ideas of authors are more clearly explained. This explanation must arrive before)*

We absolutely agree that the clash takes place between the forms with the domains that are aligned like in the orange OCP. But we cannot rule out that other clashes may occur since we don't know how the NTD moves in the active form, so maybe it collides with FRP or other OCP in a complex from different sides, leading to other clashes. The probability of the clashed configuration formation depends on the lifetime of the 1:2 complex (exactly how long dFRP stays attached to OCP after domains are aligned): if this period is shorter than one required for the second OCP to bind, than, of course, the 2:2 complex will never appear under physiological conditions. However, the principal ability of its formation helps to understand the topology of the complexes and the spatial interrelations between their constituents, which was important for our study.

In conclusion of this part: I agree that in OCPred, the CTD and FRP interaction involves FRP K102 and F76 and OCP299. By contrast, I do not agree with the idea of the clash of red OCPs when two OCPs bind to a FRP dimer. Secondly, the models shown in Figure 6A and B are right for deltaNTE OCP orange but not for WT OCPred (open domains) or OCP orange (presence of NTE, avoiding binding of FRP). Thus, if there is a clash is once the OCP becomes orange again. Why this could be useful?

We hope that our comments above were detailed enough to explain that we consider the deltaNTE OCP complex with dFRPcc as one of several intermediates presented in Figure 8 (our working model) that we managed to characterize structurally for the first time.

Then, the authors compared the FRP activities of the WT FRP dimer (that is able to monomerize) and FRPcc (oxidizing conditions) that is unable to monomerize. The experiments show that WT FRP is more effective to accelerate OCPred to OCPorange than FRPcc suggesting that the monomerization of FRP (although not essential) increases FRP effectiveness.

Figure 8: The model: The model is OK because all the possibilities are open. In the model also are more clear the ideas of the authors about how and when the second OCPred could bind the FRP dimer and

shows that the clash could occur once both OCPred were "closed" by FRP action. This is based on the model of clash shown in Figure 5E. I am still think that the binding of two OCP to an FRP dimer must be rare to occur in the cells but there is a possibility that occurs in vitro under the experimental conditions used. My idea is that after FRP dimer binding to the CTD, the interaction of the FRP monomers is weakened and the interface domain including R60,D54 and others amino acids important for FRP activity will be completely free to interact with the NTD and completely close the OCP. On the other hand, monomerization could occur after the domains will be closed (or partially closed). What it is sure that this monomerization helps the FRP activity. Nevertheless, the authors can have different ideas and prefer the "clash" hypothesis. All the time that they do not say that they demonstrated this hypothesis is OK for me.

We have also added the discussion about the potential secondary binding site, involving the NTD and the concave evolutionary conserved surface of the FRP dimer harboring several residues important for the FRP activity including R60. Thanks to the reviewer's suggestions, we have started to plan future experiments to show the possibility of spontaneous FRP monomerization focusing on exclusively OCP-CTD, however, this will not be straightforward because individual CTDs were shown to dimerize regardless of the presence of carotenoid (Moldenhauer et al. Photos Res. 2017, Muzzopappa et al Plant Physiol 2017).

In conclusion, this work present interesting results that allow a better understanding of FRP-OCP interaction and FRP activity. It is an important work that only need some more clear figures and more early and clear explanations to reinforce the message. Please, also add the experiments with the red OCPs interacting with FRP. If there is a reason why these experiments cannot be done or they will not give more information, please explain.

I hope that my comments will help you to improve your article that presents important findings and ideas that will largely influence the thinking of the field. I think that changing a little the presentation of figures and explaining better your ideas you will convince everybody that FRP monomerizes and which are the amino acids that are involved in the FRP-OCP interaction. About the mechanism of monomerization, I am less convinced but your results allow you to propose that monomerization can occurs once the OCP is again "close". All the time that this remains as a working model, it is OK for me.

Diana Kirilovsky

We would like to thank the reviewer one more time for the deep and thoughtful evaluation of our work that helped us to significantly improve the paper.

Reviewer #2 (Remarks to the Author):

In their manuscript "Topology of the OCP/FRP complexes explains the molecular mechanism controlling high light tolerance in cyanobacteria" Sluchanko et al. characterize the interactions of the fluorescence recovery protein FCP with orange carotenoid protein OCP using a variety of biophysical methods, including SAXS.

They could derive a SAXS based model of the Δ NTE(O)/ocFRPcc complex which explains the absence of 2:2 complexes due to steric hindrance and implies a mechanism for the creation of 1:1 complexes in the case FRPwt. This models is also in good agreement with crosslinking mass-spectrometry results.

Surprisingly, the authors seem to have calculated the theoretical SAXS curves of the models with CRY SOL instead directly using the fits created by CORAL. If there was a discrepancy between these curves, it should be addressed.

We would like to thank the reviewer for the careful evaluation of our work from structural biology perspective, which helped us to take into account several points that were disregarded in the original version (by our mistake). Concerning the validation of the CORAL models, we calculated fits using CRY SOL over the whole range of scattering data (and using more advanced parameters than are used by default in CORAL, see a new Table 2), while during the CORAL runs only data at low angles (most important determinants of the size/shape of a macromolecule) were considered to minimize the effect of the high-angle data (less important determinants of the size/shape of a macromolecule). Anyway, there were no significant differences in the numbers to be addressed. The ranges and all associated parameters are now indicated in a new Table 2 and Methods. Please see also below.

Despite listing the most important parameters (including the shape of the Kratky plot), the presentation of the SAXS results does not follow the guidelines for SAS based modeling of macromolecules by the IUCr (<http://journals.iucr.org/d/issues/2017/09/00/jc5010/>).

We are sincerely grateful to the reviewer for bringing this to our attention. We have now done a lot of work to get the reported parameters into a compliance with the IUCr guidelines, this analysis allowed us (and hopefully, a reader) to more deeply understand the reliability of our data. Moreover, the models and fits have now been deposited with the SASBDB under the accession codes that are now found in Table 2. The data for the L49 variant, however, which were not used for modeling and any interpretations, were moved to the Supplement in the reduced form.

It would be of particular interest to see the residuals of the fits in figures 5 and S3

Have now been added to all the plots.

as well as the pair distribution functions used for GASBOR modeling, as well as the number of beads used for fitting.

Have now been added to the revised version. In fact, plotting of $p(r)$ function for the complex and its individual components (now Fig. 5c) significantly helped us to visualize the differences in the particles involved and facilitated understanding of the SAXS data. Thank you very much for this useful suggestion! The number of beads is also added.

For the Mass based on the Porod volume a citation for the proportionality factor needs to be provided.

Corrected.

Citations are also missing for the Vc and MoW methods to derive the mass.

Corrected.

On page 20, line 650 the following citation is missing:

Blanchet, C.E., Spilotros, A., Schwemmer, F., Graewert, M.A., Kikhney, A.G., Jeffries, C.M., Franke, D., Mark, D., Zengerle, R., Cipriani, F., Fiedler, S., Roessle, M. and Svergun, D.I (2015)

Versatile sample environments and automation for biological solution X-ray scattering experiments at the P12 beamline (PETRA III, DESY) J. Appl. Cryst. 48(2)

Corrected with apologies.

Finally, I have a few more “linguistic” comments: I sincerely hope that the authors did not measure any hydrodynamic parameters in their SAXS experiment as that would imply that wither the protein complex affected the flow of the sample of the flow of the sample affected the complex. “Structural parameters in solution” would be more accurate.

Corrected.

And “Rg (reverse)” in the SAXS-tables should read “Rg (reciprocal)”.

Corrected.

Reviewer #3 (Remarks to the Author):

This paper describes the combination of genetical engineering of FRP samples of specific oligomerization state with various biophysical and biochemical techniques in order to unravel the mechanism of FRP binding to OCP. The latter is an important process in cyanobacterial photoprotection. The study shows that OCP preferentially binds dimeric FRP, which only subsequently monomerizes. The results allow for identification of potential FRP binding sites. The results are new and original and significantly advance knowledge in a dynamic field of high scientific interest. Therefore, it is of interest to a broad readership.

We are sincerely grateful to the reviewer for the kind evaluation of our results.

The paper can be recommended for publication, if the following minor issues can be resolved:

1.) page 3: ".. permitting rational engineering of the oligomeric state." If the presence of the dimeric form was already shown by SAXS, what type of engineering is necessary at this point? If this is not meant, the phrase "oligomeric state" should be more properly defined.

Corrected.

2.) page 3, bottom: the statement "Under these conditions..." is unclear. Before, the authors talk about a monomeric form, in what follows about the dimeric form. Which similar conditions are described or inferred here?

Corrected.

3.) page 4, top: "at highest protein concentration": A dependence on concentration has not been discussed up to this point. It has to be specified, what "highest protein concentration" is.

Corrected.

4.) page 6, bottom: "The SAXS profiles... were averaged and the resulting rather noisy curve...": It can only be deduced by coincidental reading of the supplementary material that 4 data sets of supposedly identical samples were averaged plus one data set of a different concentration. First, the main text should be consistent in itself and readable without the supplementary material. Second, the procedure is not understandable and seems to suggest a heterogeneity among the samples. How can this be explained?

The procedure of merging low and high concentration data is pretty common and is done in PRIMUS program (ATSAS package) to avoid heterogeneity and interparticle interference, i.e. too much particle attraction and too much particle repulsion associated with increasing solute concentrations (affecting the behavior of the SAXS profiles at low angles) at an increased S/N ratios necessary for modeling and reliable interpretations of the data at high angles. So, most of the times high angle data (the main part of the SAXS profile) are taken from the concentrated solution in a combination with the low angle data from a diluted solution to ensure that individual particles and not their chaotic associates are analyzed. This is now briefly added to the methods section and the footnote to a new Table 2 (main text) with all structural parameters derived from SAXS.

5.) page 7, middle: The form OCP^{AA} has not been introduced beyond the entry in the list of abbreviations. The reader is forced to guess that it is an equivalent of the active form OCPR.

Corrected. We have added the explanation to the introduction and paragraph preceding the mentioned one to facilitate understanding.

6.) page 7, middle: "Carotenoid-bound ... CTD" sounds wrong

We cannot agree, the ability of individual OCP CTD to bind carotenoids and dimerize was shown for the first time in our work (Moldenhauer et al Photos Res 2017) and then in other labs (Muzzopappa et al Plant Physiol 2017). A brief description of the CTD dimers binding carotenoids is now added to the text.

7.) page 10/11: the major question along with this manuscript is, why the complex topology was studied only using the deltaNTE-form of OCP, if OCP^{AA} was available and seems to resemble the active state OCPR ?

Thank you very much for this important question. This was thoroughly answered to the reviewer 1 asking almost the same question. The explanations are provided in the revised MS.

8.) The study mainly relies on SAXS data. Can the authors exclude radiation damage and why?

The doses used were adjusted to minimize the radiation damage, along with the utilization of 3% glycerol solution. In fact, a series of frames collected showed no changes that could be attributed to radiation damage. The corresponding statement is now added to the Methods section.

Minor comments:

page 3, middle: correct ".. two of its homologs..."

Corrected.

page 7, middle: "whereas already bound FRP monomer...", article seems to be missing

Corrected.

We hope that these amendments will make our revised version acceptable for publication in ***Nature Communications***.

Sincerely yours,

Nikolai N. Sluchanko, Ph.D.,

A.N. Bach Institute of Biochemistry, Research Center of Biotechnology of the Russian Academy of Sciences, 119071 Moscow, Russia

nikolai.sluchanko@mail.ru

REVIEWERS' COMMENTS:

Reviewer #1 (Remarks to the Author):

Dear Authors,

Thanks for the answer to my questions and for the new version of the article. Your answers convinced me and the new version is improved and it can be accepted to publication.

I will still ask you only two small changes in the abstract and one in the introduction:

1) The first sentence of the abstract say "High light triggers binding of the OCP to antennae complexes". This can be interpreted as that light is needed for binding that it is not the case. Please change to something saying that "High light triggers photoactivation of OCP"

2) Please finish the sentence beginning by "Unable to tightly bind....." after "1:1 complexes". And then begin the next sentence by "This could be facilitated by a transient..... ". Your data showed that this is possible and suggest that it can occur in vivo but there is no definitive demonstration.

3) In the introduction page 3, line 56, you cite articles talking about the possible site of interaction between the OCP and the core of the phycobilisome. However, your sentence talks about the fact that only OCPred interacts with phycobilisomes. This was demonstrated in Wilson et al, 2008 and Gwizdala et al, 2011. Please add

4) Attention! There is no Fig 7D. please correct the text page 15, line 382

A suggestion for authors and editor. I think that say that this work "explains" the molecular mechanism controlling high light tolerance in cyanobacteria is too strong. It is clear that it is an important work to further understand the activity of FRP in the recovery process. However, there are still a lot of open questions.

**Congratulations for this excellent work.
Diana Kirilovsky**

Reviewer #2 (Remarks to the Author):

With the exception of one "hydrodynamic property" which managed to survive in line 183 of the manuscript, the presentation and interpretation of the SAXS data are now perfectly fine for publication.

Dear reviewers,

Thank you very much for the feedback on the revised version of our manuscript. We have now introduced all the requested changes (also answered point-by-point below, see black text).

REVIEWERS' COMMENTS:

Reviewer #1 (Remarks to the Author):

Dear Authors,

Thanks for the answer to my questions and for the new version of the article. Your answers convinced me and the new version is improved and it can be accepted to publication.

Thank you!

I will still ask you only two small changes in the abstract and one in the introduction:

1) The first sentence of the abstract say "High light triggers binding of the OCP to antennae complexes". This can be interpreted as that light is needed for binding that it is not the case. Please change to something saying that "High light triggers photoactivation of OCP"

Corrected to the following:

"In cyanobacteria, high light photoactivates the Orange Carotenoid Protein (OCP) that binds to antennae complexes, dissipating energy and preventing the destruction of the photosynthetic apparatus."

2) Please finish the sentence beginning by "Unable to tightly bind....." after "1:1 complexes". And then begin the next sentence by "This could be facilitated by a transient..... ". Your data showed that this is possible and suggest that it can occur in vivo but there is no definitive demonstration.

We agree to introduce the requested change.

3) In the introduction page 3, line 56, you cite articles talking about the possible site of interaction between the OCP and the core of the phycobilisome. However, your sentence talks about the fact that only OCPred interacts with phycobilisomes. This was demonstrated in Wilson et al, 2008 and Gwizdala et al, 2011. Please add

Of course. We have added the references to this context.

4) Attention! There is no Fig 7D. please correct the text page 15, line 382

Corrected.

A suggestion for authors and editor. I think that say that this work "explains" the molecular mechanism controlling high light tolerance in cyanobacteria is too strong. It is clear that it is an important work to further understand the activity of FRP in the recovery process. However, there are still a lot of open questions.

Corrected to:

“OCP–FRP protein complex topologies suggest a mechanism for controlling high light tolerance in cyanobacteria”

Congratulations for this excellent work.
Diana Kirilovsky

Reviewer #2 (Remarks to the Author):

With the exception of one "hydrodynamic property" which managed to survive in line 183 of the manuscript, the presentation and interpretation of the SAXS data are now perfectly fine for publication.

Corrected.

Sincerely yours,
On behalf of all authors

Nikolai N. Sluchanko, PhD
Senior Researcher, Group Leader
A.N. Bach Institute of Biochemistry
Research Center of Biotechnology of the Russian Academy of Sciences
119071 Moscow, Russia
Nikolai.Sluchanko@mail.ru
+7-926-539-75-89